# On skip connections and normalisation layers in deep optimisation

**Lachlan E. MacDonald**[*]
Mathematical Institute for Data Science
Johns Hopkins University
`lemacdonald@protonmail.com`

**Jack Valmadre**
Australian Institute for Machine Learning
University of Adelaide

**Hemanth Saratchandran**
Australian Institute for Machine Learning
University of Adelaide

**Simon Lucey**
Australian Institute for Machine Learning
University of Adelaide

## Abstract

We introduce a general theoretical framework, designed for the study of gradient optimisation of deep neural networks, that encompasses ubiquitous architecture choices including batch normalisation, weight normalisation and skip connections. Our framework determines the curvature and regularity properties of multilayer loss landscapes in terms of their constituent layers, thereby elucidating the roles played by normalisation layers and skip connections in globalising these properties. We then demonstrate the utility of this framework in two respects. First, we give the only proof of which we are aware that a class of deep neural networks can be trained using gradient descent to global optima even when such optima only exist at infinity, as is the case for the cross-entropy cost. Second, we identify a novel causal mechanism by which skip connections accelerate training, which we verify predictively with ResNets on MNIST, CIFAR10, CIFAR100 and ImageNet.

## 1 Introduction

Deep, overparameterised neural networks are efficiently trainable to global optima using simple first order methods. That this is true is immensely surprising from a theoretical perspective: modern datasets and deep neural network architectures are so complex and irregular that they are essentially opaque from the perspective of classical (convex) optimisation theory. A recent surge in inspired theoretical works [22, 14, 13, 1, 52, 51, 36, 28, 29, 33, 32] has elucidated this phenomenon, showing linear convergence of gradient descent on certain classes of neural networks, with certain cost functions, to global optima. The formal principles underlying these works are identical. By taking width sufficiently large, one guarantees uniform bounds on *curvature* (via a Lipschitz gradients-type property) and *regularity* (via a Polyak-Łojasiewicz-type inequality) in a neighbourhood of initialisation. Convergence to a global optimum in that neighbourhood then follows from a well-known chain of estimates [23].

Despite significant progress, the theory of deep learning optimisation extant in the literature presents at least three significant shortcomings:

---

[*]Most done while at the Australian Institute for Machine Learning, University of Adelaide

1. *It lacks a formal framework in which to compare common practical architecture choices.* Indeed, none of the aforementioned works consider the impact of ubiquitous (weight/batch) normalisation layers. Moreover, where common architectural modifications such as skip connections *are* studied, it is unclear exactly what impact they have on optimisation. For instance, while in [13] it is shown that skip connections enable convergence with width polynomial in the number of layers, as compared with exponential width for chain networks, in [1] polynomial width is shown to be sufficient for convergence of both chain and residual networks.

2. *It lacks theoretical flexibility.* The consistent use of *uniform* curvature and regularity bounds are insufficiently flexible to enable optimisation guarantees too far away from initialisation, where the local, uniform bounds used in previous theory no longer hold. In particular, proving globally optimal convergence for deep neural nets with the cross-entropy cost was (until now) an open problem [5].

3. *It lacks practical utility.* Although it is presently unreasonable to demand quantitatively predictive bounds on practical performance, existing optimisation theory has been largely unable to inform architecture design even qualitatively. This is in part due to the first item, since practical architectures typically differ substantially from those considered for theoretical purposes.

Our purpose in this article is to take a step in addressing these shortcomings. Specifically:

1. We provide a formal framework, inspired by [45], for the study of multilayer optimisation. Our framework is sufficiently general to include all commonly used neural network layers, and contains formal results relating the curvature and regularity properties of multilayer loss landscapes to those of their constituent layers. As instances, we prove novel results on the *global* curvature and regularity properties enabled by normalisation layers and skip connections respectively, in contrast to the *local* bounds provided in previous work.

2. Using these novel, global bounds, we identify a class of weight-normalised residual networks for which, given a linear independence assumption on the data, gradient descent can be provably trained to a global optimum arbitrarily far away from initialisation. From a regularity perspective, our analysis is *strictly more flexible* than the uniform analysis considered in previous works, and in particular solves the open problem of proving global optimality for the training of deep nets with the cross-entropy cost.

3. Using our theoretical insight that skip connections aid loss regularity, we conduct a systematic empirical analysis of singular value distributions of layer Jacobians for practical layers. We are thereby able to predict that simple modifications to the classic ResNet architecture [20] will improve training speed. We verify our predictions on MNIST, CIFAR10, CIFAR100 and ImageNet.

## 2 Background

In this section we give a summary of the principles underlying recent theoretical advances in neural network optimisation. We discuss related works *after* this summary for greater clarity.

### 2.1 Smoothness and the PŁ-inequality

Gradient descent on a possibly non-convex function $\ell : \mathbb{R}^p \to \mathbb{R}_{\geq 0}$ can be guaranteed to converge to a global optimum by insisting that $\ell$ have *Lipschitz gradients* and satisfy the *Polyak-Łojasiewicz inequality*. We recall these well-known properties here for convenience.

**Definition 2.1.** Let $\beta > 0$. A continuously differentiable function $\ell : \mathbb{R}^p \to \mathbb{R}$ is said to have $\beta$-**Lipschitz gradients**, or is said to be $\beta$-**smooth** over a set $S \subset \mathbb{R}^p$ if the vector field $\nabla \ell : \mathbb{R}^p \to \mathbb{R}^p$ is $\beta$-Lipschitz. If $S$ is convex, $\ell$ having $\beta$-Lipschitz gradients implies that the inequality

$$\ell(\theta_2) - \ell(\theta_1) \leq \nabla\ell(\theta_1)^T(\theta_2 - \theta_1) + \frac{\beta}{2}\|\theta_2 - \theta_1\|^2 \tag{1}$$

holds for all $\theta_1, \theta_2 \in S$.

The $\beta$-smoothness of $\ell$ over $S$ can be thought of as a uniform bound on the *curvature* of $\ell$ over $S$: if $\ell$ is twice continuously differentiable, then it has Lipschitz gradients over any compact set $K$ with (possibly loose) Lipschitz constant given by

$$\beta := \sup_{\theta \in K} \|D^2\ell(\theta)\|, \tag{2}$$

where $D^2\ell$ is the Hessian and $\|\cdot\|$ denotes any matrix norm.

**Definition 2.2.** Let $\mu > 0$. A differentiable function $\ell : \mathbb{R}^p \to \mathbb{R}_{\geq 0}$ is said to satisfy the $\mu$-**Polyak-Łojasiewicz inequality**, or is said to be $\mu$-PŁ over a set $S \subset \mathbb{R}^p$ if

$$\|\nabla \ell(\theta)\|^2 \geq \mu \left( \ell(\theta) - \inf_{\theta' \in S} \ell(\theta') \right) \tag{3}$$

for all $\theta \in S$.

The PŁ condition on $\ell$ over $S$ is a uniform guarantee of *regularity*, which implies that all critical points of $\ell$ over $S$ are $S$-global minima; however, such a function *need not be convex*. Synthesising these definitions leads easily to the following result (cf. Theorem 1 of [23]).

**Theorem 2.3.** *Let $\ell : \mathbb{R}^p \to \mathbb{R}_{\geq 0}$ be a continuously differentiable function that is $\beta$-smooth and $\mu$-PŁ over a convex set $S$. Suppose that $\theta_0 \in S$ and let $\{\theta_t\}_{t=0}^{\infty}$ be the trajectory taken by gradient descent, with step size $\eta < 2\beta^{-1}$, starting at $\theta_0$. If $\{\theta_t\}_{t=0}^{\infty} \subset S$, then $\ell(\theta_t)$ converges to an $S$-global minimum $\ell^*$ at a linear rate:*

$$\ell(\theta_t) - \ell^* \leq \left( 1 - \mu\eta \left( 1 - \frac{\beta\eta}{2} \right) \right)^t \left( \ell(\theta_0) - \ell^* \right) \tag{4}$$

*for all $t \in \mathbb{N}$.* $\qquad\square$

Essentially, while the Lipschitz constant of the gradients controls whether or not gradient descent with a given step size can be guaranteed to *decrease* the loss at each step, the PŁ constant determines *by how much* the loss will decrease. These ideas can be applied to the optimisation of deep neural nets as follows.

## 2.2 Application to model optimisation

The above theory can be applied to parameterised models in the following fashion. Let $f : \mathbb{R}^p \times \mathbb{R}^{d_0} \to \mathbb{R}^{d_L}$ be a differentiable, $\mathbb{R}^p$-parameterised family of functions $\mathbb{R}^{d_0} \to \mathbb{R}^{d_L}$ (in later sections, $L$ will denote the number of layers of a deep neural network). Given $N$ training data $\{(x_i, y_i)\}_{i=1}^{N} \subset \mathbb{R}^{d_0} \times \mathbb{R}^{d_L}$, let $F : \mathbb{R}^p \to \mathbb{R}^{d_L \times N}$ be the corresponding parameter-function map defined by

$$F(\theta)_i := f(\theta, x_i). \tag{5}$$

Any differentiable cost function $c : \mathbb{R}^{d_L} \times \mathbb{R}^{d_L} \to \mathbb{R}_{\geq 0}$, convex in the first variable, extends to a differentiable, convex function $\gamma : \mathbb{R}^{d_L \times N} \to \mathbb{R}_{\geq 0}$ defined by

$$\gamma\left( (z_i)_{i=1}^{N} \right) := \frac{1}{N} \sum_{i=1}^{N} c(z_i, y_i), \tag{6}$$

and one is then concerned with the optimisation of the composite $\ell := \gamma \circ F : \mathbb{R}^p \to \mathbb{R}_{\geq 0}$ via gradient descent.

To apply Theorem 2.3, one needs to determine the smoothness and regularity properties of $\ell$. By the chain rule, the former can be determined given sufficient conditions on the derivatives $D\gamma \circ F$ and $DF$ (cf. Lemma 2 of [45]). The latter can be bounded by Lemma 3 of [45], which we recall below and prove in the appendix for the reader's convenience.

**Theorem 2.4.** *Let $S \subset \mathbb{R}^p$ be a set. Suppose that $\gamma : \mathbb{R}^{d_L \times N} \to \mathbb{R}_{\geq 0}$ is $\mu$-PŁ over $F(S)$ with minimum $\gamma_S^*$. Let $\lambda(DF(\theta))$ denote the smallest eigenvalue of $DF(\theta)DF(\theta)^T$. Then*

$$\|\nabla \ell(\theta)\|^2 \geq \mu \, \lambda(DF(\theta)) \left( \ell(\theta) - \gamma_S^* \right) \tag{7}$$

*for all $\theta \in S$.* $\qquad\square$

Note that Theorem 2.4 is vacuous ($\lambda(DF(\theta)) = 0$ for all $\theta$) unless in the overparameterised regime ($p \geq d_L N$). Even in this regime, however, Theorem 2.4 does not imply that $\ell$ is PŁ unless $\lambda(\theta)$ can be uniformly lower bounded by a positive constant over $S$. Although universally utilised in previous literature, such a uniform lower bound will not be possible in our *global* analysis, and our convergence theorem *does not* follow from Theorem 2.3, in contrast to previous work. Our theorem requires additional argumentation, which we believe may be of independent utility.

# 3 Related works

Convergence theorems for deep *linear* networks with the square cost are considered in [4, 2]. In [22], it is proved that the tangent kernel of a multi-layer perceptron (MLP) becomes approximately constant over all of parameter space as width goes to infinity, and is positive-definite for certain data distributions, which by Theorem 2.4 implies that all critical points are global minima. Strictly speaking, however, [22] does not prove convergence of gradient descent: the authors consider only gradient flow, and leave Lipschitz concerns untouched. The papers [14, 13, 1, 52, 51, 36] prove that overparameterized neural nets of varying architectures can be optimised to global minima *close to initialisation* by assuming sufficient width of several layers. While [1] does consider the cross-entropy cost, convergence to a global optimum is *not* proved: it is instead shown that perfect classification accuracy can be achieved close to initialisation during training. Improvements on these works have been made in [33, 32, 7], wherein large width is required of only a single layer.

It is identified in [28] that linearity of the final layer is key in establishing the approximate constancy of the tangent kernel for wide networks that was used in [14, 13]. By making explicit the implicit use of the PŁ condition present in previous works [14, 13, 1, 52, 51, 36], [29] proves a convergence theorem even with *nonlinear* output layers. The theory explicated in [29] is formalised and generalised in [45]. A key weakness of all of the works mentioned thus far (bar the purely formal [45]) is that their hypotheses imply that optimisation trajectories are always close to initialisation. Without this, there is no obvious way to guarantee the PŁ-inequality along the optimisation trajectory, and hence no way to guarantee one does not converge to a suboptimal critical point. However such training is not possible with the cross-entropy cost, whose global minima only exist at infinity. There is also evidence to suggest that such training must be avoided for state-of-the-art test performance [8, 15, 26]. In contrast, our theory gives convergence guarantees even for trajectories that travel arbitrarily far from initialisation, and is the only work of which we are aware that can make this claim.

Among the tools that make our theory work are skip connections [20] and weight normalisation [41]. The smoothness properties of normalisation schemes have previously been studied [42, 40], however they only give pointwise estimates comparing normalised to non-normalised layers, and do not provide a global analysis of Lipschitz properties as we do. The regularising effect of skip connections on the loss landscape has previously been studied in [35], however this study is not tightly linked to optimisation theory. Skip connections have also been shown to enable the interpretation of a neural network as coordinate transformations of data manifolds [19]. Mean field analyses of skip connections have been conducted in [49, 30] which necessitate large width; our own analysis does not. A similarly general framework to that which we supply is given in [47, 48]; while both encapsulate all presently used architectures, that of [47, 48] is designed for the study of infinite-width tangent kernels, while ours is designed specifically for optimisation theory. Our empirical singular value analysis of skip connections complements existing theoretical work using random matrix theory [18, 37, 39, 34, 16]. These works have not yet considered the shifting effect of skip connections on layer Jacobians that we observe empirically.

Our theory also links nicely to the intuitive notions of gradient propagation [20] and dynamical isometry already present in the literature. In tying Jacobian singular values rigorously to loss regularity in the sense of the Polyak-Lojasiewicz inequality, our theory provides a new link between dynamical isometry and optimisation theory [43, 38, 46]: specifically dynamical isometry ensures better PŁ conditioning and therefore faster and more reliable convergence to global minima. In linking this productive section of the literature to optimisation theory, our work may open up new possibilities for convergence proofs in the optimisation theory of deep networks. We leave further exploration of this topic to future work.

Due to this relationship with the notion of dynamical isometry, our work also provides optimisation-theoretic support for the empirical analyses of [24, 6] that study the importance of layerwise dynamical isometry for trainability and neural architecture search [27, 44, 25]. Recent work on deep kernel shaping shows via careful tuning of initialisation and activation functions that while skip connections and normalisation layers may be *sufficient* for good trainability, they are not *necessary* [31, 50]. Other recent work has also shown benefits to inference performance by removing skip connections from the trained model using a parameter transformation [11] or by removing them from the model altogether and incorporating them only into the optimiser [12].

Finally, a line of work has recently emerged on training in the more realistic, large learning rate regime known as "edge of stability" [9, 3, 10]. This intriguing line of work diverges from ours, and its integration into the framework we present is a promising future research direction.

# 4    Formal framework

In this section we will define our theoretical framework. We will use $\mathbb{R}^{n \times m}$ to denote the space of $n \times m$ matrices, and vectorise rows first. We use $\mathrm{Id}_m$ to denote the $m \times m$ identity matrix, and $1_n$ to denote the vector of ones in $\mathbb{R}^n$. We use $\otimes$ to denote the Kronecker (tensor) product of matrices, and given a vector $v \in \mathbb{R}^n$ we use $\mathrm{diag}(v)$ to denote the $n \times n$ matrix whose leading diagonal is $v$. Given a matrix $A \in \mathbb{R}^{n \times m}$, and a seminorm $\|\cdot\|$ on $\mathbb{R}^m$, $\|A\|_{row}$ will be used to denote the vector of $\|\cdot\|$-seminorms of each row of $A$. The smallest singular value of $A$ is denoted $\sigma(A)$, and the smallest eigenvalue of $AA^T$ denoted $\lambda(A)$. Full proofs of all of our results can be found in the appendix.

Our theory is derived from the following formal generalisation of a deep neural network.

**Definition 4.1.** By a **multilayer parameterised system (MPS)** we mean a family $\{f_l : \mathbb{R}^{p_l} \times \mathbb{R}^{d_{l-1} \times N} \to \mathbb{R}^{d_l \times N}\}_{l=1}^L$ of functions. Given a data matrix $X \in \mathbb{R}^{d_0 \times N}$, we denote by $F : \mathbb{R}^{\sum_{i=1}^L p_i} \to \mathbb{R}^{d_L \times N}$ the **parameter-function map**[2] defined by

$$F(\vec{\theta}) := f_L(\theta_L) \circ \cdots \circ f_1(\theta_1)(X), \tag{8}$$

for $\vec{\theta} = (\theta_1, \ldots, \theta_L)^T \in \mathbb{R}^{\sum_{i=1}^L p_i}$.

Definition 4.1 is sufficiently general to encompass all presently used neural network layers, but *we will assume without further comment from here on in that all layers are continuously differentiable.* Before we give examples, we record the following result giving the form of the derivative of the parameter-function map, which follows easily from the chain rule. It will play a key role in the analysis of the following subsections.

**Proposition 4.2.** *Let* $\{f_l : \mathbb{R}^{p_l} \times \mathbb{R}^{d_{l-1} \times N} \to \mathbb{R}^{d_l \times N}\}_{l=1}^L$ *be a MPS, and* $X \in \mathbb{R}^{d_0 \times N}$ *a data matrix. For* $1 \le l \le L$, *denote the derivatives of* $f_l$ *with respect to the* $\mathbb{R}^{p_l}$ *and* $\mathbb{R}^{d_{l-1}}$ *variables by* $Df_l$ *and* $Jf_l$ *respectively, and let* $f_{<l}(\vec{\theta}, X)$ *denote the composite*

$$f_{<l}(\vec{\theta}, X) := f_{l-1}(\theta_{l-1}) \circ \cdots \circ f_1(\theta_1)(X). \tag{9}$$

*The derivative* $DF$ *of the associated parameter-function map is given by*

$$DF(\vec{\theta}) = \left( D_{\theta_1} F(\vec{\theta}), \ldots, D_{\theta_L} F(\vec{\theta}) \right), \tag{10}$$

*where for* $1 \le l < L$, $D_{\theta_l} F(\vec{\theta})$ *is given by the formula*

$$\left( \prod_{j=l+1}^L Jf_j\big(\theta_j, f_{<j}(\vec{\theta}, X)\big) \right) Df_l\big(\theta_l, f_{<l}(\vec{\theta}, X)\big), \tag{11}$$

*with the product taken so that indices are arranged in descending order from left to right.* □

All common differentiable neural network layers fit into this framework; we record some examples in detail in the appendix. We will see in the next section that insofar as one wishes to guarantee global smoothness, the usual parameterisation of affine layers is poor, although this defect can be ameliorated to differing extents by normalisation strategies.

## 4.1    Smoothness

In this subsection, we give sufficient conditions for the derivative of the parameter-function map of a MPS to be bounded and Lipschitz. We are thereby able to give sufficient conditions for any associated loss function to have Lipschitz gradients on its sublevel sets. We begin with a formal proposition that describes the Lipschitz properties of the derivative of the parameter-function map of a MPS in terms of those of its constituent layers.

---

[2]$\mathbb{R}^{d_L \times N}$ is canonically isomorphic to the space of $\mathbb{R}^{d_L}$-valued functions on an $N$-point set.

**Proposition 4.3.** *Let $\{f_l : \mathbb{R}^{p_l} \times \mathbb{R}^{d_{l-1} \times N} \to \mathbb{R}^{d_l \times N}\}_{l=1}^{L}$, and let $\{S_l \subset \mathbb{R}^{p_l}\}_{l=1}^{L}$ be subsets of the parameter spaces. Suppose that for each bounded set $B_l \subset \mathbb{R}^{d_{l-1} \times N}$, the maps $f_l$, $Df_l$ and $Jf_l$ are all bounded and Lipschitz on $S_l \times B_l$. Then for any data matrix $X \in \mathbb{R}^{d_0 \times N}$, the derivative $DF$ of the associated parameter-function map $F : \mathbb{R}^p \to \mathbb{R}^{d_L \times N}$ is bounded and Lipschitz on $S := \prod_{j=1}^{L} S_j \subset \mathbb{R}^p$.* □

Proposition 4.3 has the following immediate corollary, whose proof follows easily from Lemma B.1.

**Corollary 4.4.** *Let $c : \mathbb{R}^{d_L} \to \mathbb{R}$ be any cost function whose gradient is bounded and Lipschitz on its sublevel sets $Z_\alpha := \{z \in \mathbb{R}^{d_L} : c(z) \leq \alpha\}$. If $\{f_l : \mathbb{R}^{p_l} \times \mathbb{R}^{d_{l-1} \times N} \to \mathbb{R}^{d_l \times N}\}_{l=1}^{L}$ is any MPS satisfying the hypotheses of Proposition 4.3 and $X \in \mathbb{R}^{d_0 \times N}$, then the associated loss function $\ell := \gamma \circ F$ has Lipschitz gradients over $S := \prod_{l=1}^{L} S_l$.* □

The most ubiquitous cost functions presently in use (the mean square error and cross entropy functions) satisfy the hypotheses of Corollary 4.4. We now turn to an analysis of common layer types in deep neural networks and indicate to what extent they satisfy the hypotheses of Proposition 4.3.

**Theorem 4.5.** *Fix $\epsilon > 0$. The following layers satisfy the hypotheses of Proposition 4.3 over all of parameter space.*

*1. Continuously differentiable nonlinearities.*
*2. Bias-free $\epsilon$-weight-normalised or $\epsilon$-entry-normalised affine layers[3].*
*3. Any residual block whose branch is a composite of any of the above layer types.*

*Consequently, the loss function of any neural network composed of layers as above, trained with a cost function satisfying the hypotheses of Corollary 4.4, has globally Lipschitz gradients along any sublevel set.* □

The proof we give of Theorem 4.5 also considers composites bn ∘ aff of batch norm layers with affine layers. Such composites satisfy the hypotheses of Proposition 4.3 only over sets in data-space $\mathbb{R}^{d \times N}$ which consist of matrices with nondegenerate covariance. Since such sets are generic (probability 1 with respect to any probability measure that is absolutely continuous with respect to Lebesgue measure), batch norm layers satisfy the hypotheses of Proposition 4.3 with high probability over random initialisation.

Theorem 4.5 says that normalisation of parameters enables *global* analysis of the loss, while standard affine layers, due to their unboundedness, are well-suited to analysis only over bounded sets in parameter space.

## 4.2 Regularity

Having examined the Lipschitz properties of MPS and given examples of layers with *global* Lipschitz properties, let us now do the same for the *regularity* properties of MPS. Formally one has the following simple result.

**Proposition 4.6.** *Let $\{f_l : \mathbb{R}^{p_l} \times \mathbb{R}^{d_{l-1} \times N} \to \mathbb{R}^{d_l \times N}\}_{i=1}^{L}$ be a MPS and $X \in \mathbb{R}^{d_0 \times N}$ a data matrix. Then*

$$\sum_{l=1}^{L} \lambda\big(Df_l\big(\theta_l, f_{<l}(\vec{\theta}, X)\big)\big) \prod_{j=l+1}^{L} \lambda\big(Jf_j(\theta_j, f_{<j}(\vec{\theta}, X))\big) \tag{12}$$

*is a lower bound for $\lambda\big(DF(\vec{\theta})\big)$.* □

Proposition 4.6 tells us that to guarantee good regularity, it suffices to guarantee good regularity of the constituent layers. In fact, since Equation (12) is a sum of non-negative terms, it suffices merely to guarantee good regularity of the parameter-derivative of the *first* layer[4], and of the input-output Jacobians of every *subsequent* layer. Our next theorem says that residual networks with appropriately normalised branches suffice for this.

**Theorem 4.7.** *Let $\{g_l : \mathbb{R}^{p_l} \times \mathbb{R}^{d_{l-1} \times N} \to \mathbb{R}^{d_l \times N}\}_{i=1}^{L}$ be a MPS and $X \in \mathbb{R}^{d_0 \times N}$ a data matrix for which the following hold:*

---

[3]The theorem also holds with normalised biases. We make the bias-free assumption assumption purely out of notational convenience.

[4]The parameter derivatives of higher layers are more difficult to analyse, due to nonlinearities.

1. $d_{l-1} \geq d_l$, and $\|Jg_l(\theta_l, Z)\|_2 < 1$ for all $\theta_l \in \mathbb{R}^{p_l}$, $Z \in \mathbb{R}^{d_{l-1} \times N}$ and $l \geq 2$.

2. $N \leq d_0$, $X$ is full rank, $p_1 \geq d_1 d_0$ and $f_1 : \mathbb{R}^{p_1} \times \mathbb{R}^{d_0 \times N} \to \mathbb{R}^{d_1 \times N}$ is a P-parameterised affine layer, for which $DP(\theta_1)$ is full rank for all $\theta_1 \in \mathbb{R}^{p_1}$.

*For any sequence $\{I_l : \mathbb{R}^{d_{l-1} \times N} \to \mathbb{R}^{d_l \times N}\}_{l=2}^L$ of linear maps whose singular values are all equal to 1, define a new MPS $\{f_l\}_{l=1}^L$ by $f_1 := g_1$, and $f_l(\theta_l, X) := I_l X + g_l(\theta_l, X)$. Let $F : \mathbb{R}^p \to \mathbb{R}^{d_L \times N}$ be the parameter-function map associated to $\{f_l\}_{l=1}^L$ and $X$. Then $\lambda(DF(\vec{\theta})) > 0$ (but not uniformly so) for all $\vec{\theta} \in \mathbb{R}^p$.* $\qquad\square$

In the next and final subsection we will synthesise Theorem 4.5 and Theorem 4.7 into our main result: a global convergence theorem for gradient descent on appropriately normalised residual networks.

# 5  Main theorem

We will consider *normalised residual networks*, which are MPS of the following form. The first layer is an $\epsilon$-entry-normalised, bias-free affine layer $f_1 = \text{aff}_{\text{en}} : \mathbb{R}^{d_1 \times d_0} \times \mathbb{R}^{d_0 \times N} \to \mathbb{R}^{d_1 \times N}$ (cf. Example A.1). Every subsequent layer is a residual block

$$f_l(\theta_l, X) = I_l X + g(\theta_l, X). \tag{13}$$

Here, for all $l \geq 2$, we demand that $d_{l-1} \geq d_l$, $I_i : \mathbb{R}^{d_{i-1} \times N} \to \mathbb{R}^{d_i \times N}$ is some linear map with all singular values equal to 1, and the residual branch $g(\theta_i, X)$ is a composite of weight- or entry-normalised, bias-free[5] affine layers $\text{aff}_P$ (cf. Example A.1), rescaled so that $\|P(w)\|_2 < 1$ uniformly for all parameters $w$, and elementwise nonlinearities $\Phi$ for which $\|D\phi\| \leq 1$ everywhere (Example A.2). These hypotheses ensure that Theorem 4.7 holds for $\{f_i\}_{i=1}^L$ and $X$: see the Appendix for a full proof.

We emphasise again that our main theorem below does *not* follow from the usual argumentation using smoothness and the PŁ inequality, due to the lack of a *uniform* PŁ bound. Due to the novelty of our technique, which we believe may be of wider utility where uniform regularity bounds are unavailable, we include an idea of the proof below.

**Theorem 5.1.** *Let $\{f_i\}_{i=1}^L$ be a normalised residual network; $X$ a data matrix of linearly independent data, with labels $Y$; and $c$ any continuously differentiable, convex cost function. Then there exists a learning rate $\eta > 0$ such that gradient descent on the associated loss function converges from **any** initialisation to a global minimum.*

*Idea of proof.*  One begins by showing that Theorems 4.5 and 4.7 apply to give globally $\beta$-Lipschitz gradients and a positive smallest eigenvalue of the tangent kernel at all points in parameter space. Thus for learning rate $\eta < 2\beta^{-1}$, there exists a positive sequence $(\mu_t = \mu\lambda(DF(\theta_t)))_{t \in \mathbb{N}}$ (see Theorem 2.4) such that $\|\nabla \ell_t\|^2 \geq \mu_t \ell_t$, for which the loss iterates $\ell_t$ therefore obey

$$\ell_t - \ell^* \leq \prod_{i=0}^t (1 - \mu_i \alpha)(\ell_0 - \ell^*),$$

where $\alpha = \eta(1 - 2\beta^{-1}\eta) > 0$. To show global convergence one must show that $\prod_{t=0}^\infty (1 - \mu_t \alpha) = 0$.

If $\mu_t$ can be uniformly lower bounded (e.g. for the square cost) then Theorem 2.3 applies to give convergence as in all previous works. However, $\mu_t$ *cannot* be uniformly lower bounded in general (e.g. for the cross-entropy cost). We attain the general result by showing that, despite admitting no non-trivial lower bound in general, $\mu_t$ can always be guaranteed to vanish *sufficiently slowly* that global convergence is assured. $\qquad\square$

We conclude this section by noting that practical deep learning problems typically do not satisfy the hypotheses of Theorem 5.1: frequently there are more training data than input dimensions (such as for MNIST and CIFAR), and many layers are not normalised or skip-connected. Moreover our Lipschitz bounds are worst-case, and will generally lead to learning rates much smaller than are used in practice. Our strong hypotheses are what enable a convergence guarantee from any initialisation, whereas in practical settings initialisation is of key importance. Despite the impracticality of Theorem 5.1, in the next section we show that the ideas that enable its proof nonetheless have practical implications.

---

[5]The theorem also holds with normalised biases.

# 6 Practical implications

Our main theorem is difficult to test directly, as it concerns only worst-case behaviour which is typically avoided in practical networks which do not satisfy its hypotheses. However, our framework more broadly nonetheless yields practical insights. Informally, Theorem 4.7 gives conditions under which:

> ***skip connections aid optimisation by improving loss regularity.***

In this section, we conduct an empirical analysis to demonstrate that *this insight holds true even in practical settings*, and thereby obtain a novel, *causal* understanding of the benefits of skip connections in practice. With this causal mechanism in hand, we recommend simple architecture changes to practical ResNets that consistently (albeit modestly) improve convergence speed as predicted by theory. All code is available at `https://github.com/lemacdonald/skip-connections-normalisation/`.

## 6.1 Singular value distributions

Let us again recall the setting $\ell = \gamma \circ F$ of Theorem 2.4. In the overparameterised setting, the smallest singular value of $DF$ gives a *pessimistic lower bound* on the ratio $\|\nabla\ell\|^2/(\ell - \ell^*)$, and hence a *pessimistic* lower bound on training speed (cf. Theorems 2.4, 2.3). Indeed, this lower bound is only attained when the vector $\nabla\gamma$ perfectly aligns with the smallest singular subspace of $DF$: a probabilistically unlikely occurence. In general, since $\nabla\ell = DF^T \nabla\gamma$, *the entire singular value distribution* of $DF$ at a given parameter will play a role in determining the ratio $\|\nabla\ell\|^2/(\ell - \ell^*)$, and hence training speed.

Now $DF$ is partly determined by products of layer Jacobians (cf. Proposition 4.2). As such, the distribution of singular values of $DF$ is determined in part by the distribution of singular values of such layer Jacobian products, which are themselves determined by the singular value distributions of each of the individual layer Jacobians.

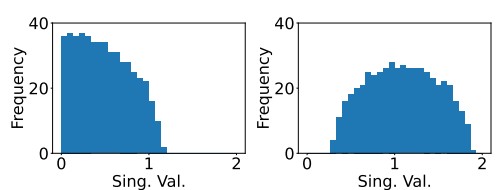

Figure 1: Singular value histogram of $500 \times 500$ matrix $A$ (left) with entries sampled iid from $U(-1/\sqrt{500}, 1/\sqrt{500})$. Adding an identity matrix (right) shifts the distribution upwards.

In particular, if through some architectural intervention each layer Jacobian could have its singular value distribution shifted upwards, we would expect the singular value distribution of $DF$ to be shifted upwards, too. Our argument in the previous paragraph then suggests that such an intervention will result in faster training, provided of course that the upwards-shifting is not so large as to cause exploding gradients.

Figure 1 shows in the *linear* setting that a skip connection constitutes precisely such an intervention. We hypothesise that this continues to hold even in the nonlinear setting.

**Hypothesis 6.1.** *The addition of a deterministic skip connection, all of whose singular values are 1, across a composite of possibly nonlinear random layers, shifts upwards the singular value distribution of the corresponding composite Jacobian[6], thereby improving convergence speed at least initially.*

We test Hypothesis 6.1 in the next subsections.

## 6.2 MNIST

Recall that the ResNet architecture [20] consists in part of a composite of residual blocks

$$f(\theta, A, X) = AX + g(\theta, X), \tag{14}$$

where $A$ is either the identity transformation, in the case when the dimension of the output of $g(\theta, X)$ is the same as the dimension of its input; or a randomly initialised 1x1 convolution otherwise.

---

[6]At least for some common initialisation schemes.

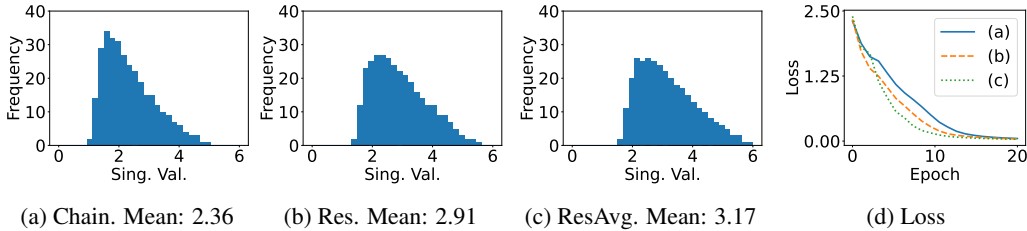

(a) Chain. Mean: 2.36    (b) Res. Mean: 2.91    (c) ResAvg. Mean: 3.17    (d) Loss

Figure 2: (a)-(c) Singular value histograms of composite layer Jacobians averaged over first 10 training iterations. Distributions are shifted upwards as deterministic skip connections are added, resulting in faster convergence (d). Means over 10 trials shown.

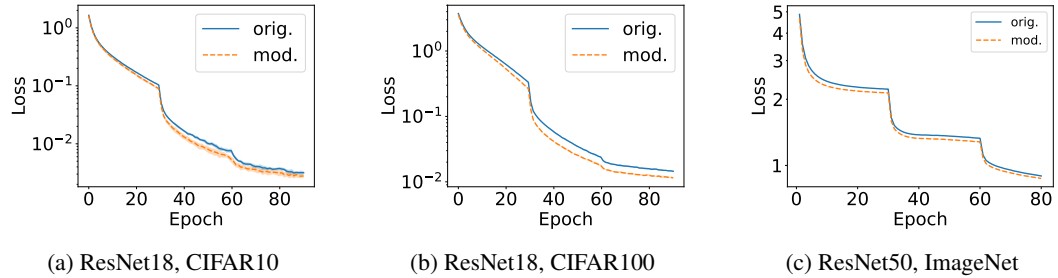

(a) ResNet18, CIFAR10    (b) ResNet18, CIFAR100    (c) ResNet50, ImageNet

Figure 3: Mean training loss curves for ResNets on CIFAR (10 trials) and ImageNet (3 trials), with 1 standard deviation shaded. The modifications improve training speed as predicted.

Hypothesis 6.1 predicts that the additions of the identity skip connections will shift upwards the singular value distributions of composite layer Jacobians relative to the equivalent chain network, thus improving convergence speed. It also predicts that replacing 1x1 convolutional skip connections $A$ with $I + A$, where $I$ is deterministic with all singular values equal to 1, will do the same.

We first test this hypothesis by doing gradient descent, with a learning rate of 0.1, on 32 randomly chosen data points from the MNIST training set[7]. We compare three models, all with identical initialisations for each trial run using the default PyTorch initialisation:

1. (Chain) A batch-normed convolutional chain network, with six convolutional layers.
2. (Res) The same as (1), but with convolutional layers 2-3 and 5-6 grouped into two residual blocks, and with an additional 1x1 convolution as the skip connection in the second residual block, as in Equation (14).
3. (ResAvg) The same as (2), but with the 1x1 convolutional skip connection of Equation (14) replaced by

$$\tilde{f}(\theta, A, X) = (I + A)X + g(\theta, X), \tag{15}$$

where $I$ is an average pool, rescaled to have all singular values equal to 1.

Hypothesis 6.1 predicts that the singular value distributions of the composite layer Jacobians will be more positively shifted going down the list, resulting in faster convergence. This is indeed what we observe (Figure 2).

## 6.3   CIFAR and ImageNet

We now test Hypothesis 6.1 on CIFAR and ImageNet. We replace all of the convolution-skip connections in the ResNet architecture [20] with sum(average pool, convolution)-skip connections as in Equation (15) above, leaving all else unchanged. Hypothesis 6.1 predicts that these modifications will improve training speed, which we verify using default PyTorch initialisation schemes.

We trained PreAct-ResNet18 on CIFAR10/100 and PreAct-ResNet50 [20] on ImageNet using standard training regimes (details can be found in the appendix), performing respectively 10 and 3 trial runs

---

[7]The input-output Jacobians of MPS layers scale with the square of the number of training points, making their computation with a large number of data points prohibitively expensive computationally.

on CIFAR and ImageNet. We performed the experiments at a range of different learning rates. We have included figures for the best performing learning rates on the *original* model (measured by loss value averaged over the final epoch) in Figure 3, with additional plots and validation accuracies in the appendix. Validation accuracies were not statistically significantly different between the two models on CIFAR10/100. Although the modified version had statistically significantly better validation accuracy in the ImageNet experiment, we believe this is only due to the faster convergence, as the training scheme was not sufficient for the model to fully converge.

## 7 Discussion

Our work suggests some open research problems. First, the recently-developed edge of stability theory [9] could be used in place of our Lipschitz bounds to more realistically characterise training of practical nets with large learning rates. Second, like in pervious works [7], the heavy-lifting for our PŁ-type bounds is all done by the parameter-derivative of a single layer, and the bounds would be significantly improved by an analysis that considers all layers. Third, extension of the theory to SGD is desirable. Fourth, the dependence of Hypothesis 6.1 on weight variance should be investigated. Fifth, our empirical results on the impact of skip connections on singular value distributions suggests future work using random matrix theory [18].

Beyond these specifics, our formal framework provides a setting in which all neural network layers can be analysed in terms of their effect on the key loss landscape properties of smoothness and regularity, and is the first to demonstrate that a *uniform* bound on regularity is not necessary to prove convergence. We hope the tools we provide in this paper will be of use in extending deep learning optimisation theory to more practical settings than has so far been the case.

## 8 Conclusion

We gave a formal theoretical framework for studying the optimisation of multilayer systems. We used the framework to give the first proof that a class of deep neural networks can be trained by gradient descent even to global optima at infinity. Our theory generates the novel insight that skip connections aid optimisation speed by improving loss regularity, which we verified empirically using practical datasets and architectures.

## 9 Acknowledgements

We thank the anonymous reviewers for their time reviewing the manuscript. Their critiques helped to improve the paper.

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

**Computational resources:** Both the exploratory and final experiments for this paper were conducted using a desktop machine with two Nvidia RTX A6000 GPUs, for a total running time of approximately 150 hours.

## A  Examples of MPS

All commonly used neural network layers fit into the framework of multilayer parameterised systems (MPS). Here we list the examples of most relevance to our work.

**Example A.1.** An *affine layer* aff : $\mathbb{R}^{d_1 \times d_0 + d_1} \times \mathbb{R}^{d_0 \times N} \to \mathbb{R}^{d_1 \times N}$ is given by the formula

$$\text{aff}(A, b, X) := AX + b1_N^T. \tag{16}$$

A routine calculation shows that

$$J\text{aff}(A, b, X) = A \otimes \text{Id}_N, \tag{17}$$

while

$$D\text{aff}(A, b, X) = \left(\text{Id}_{d_1} \otimes X^T, \text{Id}_{d_1} \otimes 1_N\right). \tag{18}$$

More generally, if $P : \mathbb{R}^p \to \mathbb{R}^{d_1 \times d_0}$ denotes any continuously differentiable map, then one obtains a *P-parameterised affine layer*

$$\text{aff}_P(w, b, X) := \text{aff}(P(w), b, X) = P(w)X + b1^T. \tag{19}$$

One has

$$J\text{aff}_P(w, b, X) = P(w) \otimes \text{Id}_N \tag{20}$$

and, by the chain rule,

$$D\text{aff}_P(w, b, X) = \left((\text{Id}_{d_1} \otimes X^T)DP(w), \text{Id}_{d_1} \otimes 1\right), \tag{21}$$

where $DP(w) \in \mathbb{R}^{d_0 N \times p}$ is the derivative of $P$ at $w$. Common examples include $\epsilon$-*weight normalisation* $\text{wn}(w) := (\epsilon + \|w\|_{row}^2)^{-\frac{1}{2}} w$ [41] and convolutions, which send convolutional kernels to associated Toeplitz matrices. We will also consider $\epsilon$-*entry normalisation* $\text{en}(w) := (\epsilon + w^2)^{-\frac{1}{2}} w$, with operations applied entrywise.

**Example A.2.** A *(parameter-free) elementwise nonlinearity* $\Phi : \mathbb{R}^{d_0 \times N} \to \mathbb{R}^{d_0 \times N}$ defined by a continuously differentiable function $\phi : \mathbb{R} \to \mathbb{R}$ is given by applying $\phi$ to every component of a matrix $X \in \mathbb{R}^{d_0 \times N}$. Extension to the parameterised case is straightforward.

**Example A.3.** A *(parameter-free) batch normalisation (BN) layer* bn : $\mathbb{R}^{d_0 \times N} \to \mathbb{R}^{d_0 \times N}$ is given by the formula

$$\text{bn}(X) := \frac{X - \mathbb{E}[X]}{\sqrt{\epsilon + \sigma[X]^2}}, \tag{22}$$

where $\epsilon > 0$ is some fixed hyperparameter and $\mathbb{E}$ and $\sigma$ denote the row-wise mean and standard deviation. The parameterised BN layer from [21], with scaling and bias parameters $\gamma$ and $\beta$ respectively, is given simply by postcomposition $\text{aff}_{\text{diag}}(\gamma, \beta, \cdot) \circ \text{bn}$ with a diag-parameterised affine layer (Example A.1).

**Example A.4.** A *residual block* $f : \mathbb{R}^p \times \mathbb{R}^{d_0 \times N} \to \mathbb{R}^{d_1 \times N}$ can be defined given any other layer (or composite thereof) $g : \mathbb{R}^p \times \mathbb{R}^{d_0 \times N} \to \mathbb{R}^{d_1 \times N}$ by the formula

$$f(\theta, X) := IX + g(\theta, X), \tag{23}$$

where $I : \mathbb{R}^{d_0 \times N} \to \mathbb{R}^{d_1 \times N}$ is some linear transformation. In practice, $I$ is frequently the identity map [20]; our main theorem will concern the case where $I$ has all singular values equal to 1.

## B  Proofs

*Proof of Theorem 2.4.* Using the fact that $D\ell = (D\gamma \circ F) \cdot DF$, we compute:

$$\begin{aligned}
\|\nabla\ell(\theta)\|^2 &= \langle DF(\theta)^T \nabla\gamma(F(\theta)), DF(\theta)^T \nabla\gamma(F(\theta))\rangle \\
&= \langle \nabla\gamma(F(\theta)), DF(\theta)DF(\theta)^T \nabla\gamma(F(\theta))\rangle \\
&\geq \lambda(DF(\theta))\|\nabla\gamma(F(\theta))\|^2 \\
&\geq \mu\lambda(DF(\theta))\Big(\gamma(F(\theta)) - \inf_{\theta'} \gamma(F(\theta'))\Big),
\end{aligned}$$

where the first inequality follows from the standard estimate $\langle v, AA^T v \rangle \geq \lambda_{min}(AA^T)\|v\|^2$, and the final inequality follows from the fact that $\gamma$ is $\mu$-PŁ over the set $\{F(\theta) : \theta \in \mathbb{R}^p\}$. $\qquad\square$

Our proof of Proposition 4.3 requires the following standard lemma.

**Lemma B.1.** *Let $\{g_i : \mathbb{R}^p \to \mathbb{R}^{m_i \times m_{i-1}}\}_{i=1}^n$ be a family of matrix-valued functions. If, with respect to some submultiplicative matrix norm, each $g_i$ is bounded by $b_i$ and Lipschitz with constant $c_i$ on a set $S \subset \mathbb{R}^p$, then their pointwise matrix product $\theta \mapsto \prod_{i=1}^n g_i(\theta)$ is also bounded and Lipschitz on $S$, with bound $\prod_{i=1}^n b_i$ and Lipschitz constant $\sum_{i=1}^n c_i\big(\prod_{j \neq i} b_j\big)$.* $\qquad\square$

*Proof of Lemma B.1.* We prove the lemma by induction. When $n = 2$, adding and subtracting a copy of $g_1(\theta)g_2(\theta')$ and using the triangle inequality implies that $\|g_1 g_2(\theta) - g_1 g_2(\theta')\|$ is bounded by

$$\|g_1(\theta)(g_2(\theta) - g_2(\theta'))\| + \|(g_1(\theta) - g_1(\theta'))g_2(\theta')\|.$$

Applying submultiplicativity of the matrix norm and the bounds provided by the $b_i$ and $c_i$ gives

$$\|g_1 g_2(\theta) - g_1 g_2(\theta')\| \leq (b_1 c_2 + b_2 c_1)\|\theta - \theta'\|.$$

Now suppose we have the result for $n = k$. Writing $\prod_{i=1}^{k+1} g_i$ as $g_1 \prod_{i=2}^{k+1} g_i$ and applying the above argument, the induction hypothesis tells us that $\prod_{i=1}^{k+1} g_i$ is indeed bounded by $\prod_{i=1}^{k+1} b_i$ and Lipschitz with Lipschitz constant $\sum_{i=1}^{k+1} c_i\big(\prod_{j \neq i} b_j\big)$. The result follows. $\qquad\square$

*Proof of Proposition 4.3.* By Proposition 4.2, it suffices to show that for each $1 \leq l \leq L$, the function

$$\vec{\theta} \mapsto \prod_{j=l+1}^L Jf_j\big(\theta_j, f_{<j}(\vec{\theta}, X)\big) Df_l\big(\theta_l, f_{<l}(\vec{\theta}, X)\big) \tag{24}$$

is bounded and Lipschitz on $S$. To show this, we must first prove that each map $\vec{\theta} \mapsto f_{<j}(\vec{\theta}, X)$ is bounded and Lipschitz on $S$. This we prove by induction.

By hypothesis, $\vec{\theta} \mapsto f_1(\vec{\theta}, X) = f_1(\theta_1, X)$ is bounded and Lipschitz on $S$. Suppose now that for $j > 1$, one has $\vec{\theta} \mapsto f_{<j}(\vec{\theta}, X)$ bounded and Lipschitz on $S$. Then the range of $S \ni \theta \mapsto f_{<j}(\vec{\theta}, X)$ is a bounded subset of $\mathbb{R}^{d_j \times N}$. By hypothesis on $f_j$, it then follows that $\theta \mapsto f_{<j+1}(\vec{\theta}, X) = f_j\big(\theta_j, f_{<j}(\vec{\theta}, X)\big)$ is bounded and Lipschitz on $S$.

The hypothesis on the $Jf_j$ and $Df_j$ now implies that the maps $\vec{\theta} \mapsto Jf_j\big(\theta_j, f_{<j}(\vec{\theta}, X)\big), l+1 \leq j \leq L$, and $\vec{\theta} \mapsto Df_l\big(\theta_l, f_{<l}(\vec{\theta}, X)\big)$ are all bounded and Lipschitz on $S$. In particular, as a product of bounded and Lipschitz functions, the map given in Equation (24) is also bounded and Lipschitz on $S$. Therefore $DF$ is bounded and Lipschitz on $S$. $\qquad\square$

*Proof of Corollary 4.4.* By hypothesis, $F(S)$ is a bounded subset of $\mathbb{R}^{d_L \times N}$. Continuity of $\gamma$ then implies that $\gamma(F(S))$ is a bounded subset of $\mathbb{R}$, so that $F(S)$ is contained in a sublevel set of $\gamma$. The result now follows from the hypotheses. $\qquad\square$

To prove Theorem 4.5 it will be convenient to recall some tensor calculus. If $f : \mathbb{R}^{n_1 \times n_2} \to \mathbb{R}^{m_1 \times m_2}$ is a matrix-valued, differentiable function of a matrix-valued variable, its derivative $Df$ can be regarded as a map $\mathbb{R}^{n_1 \times n_2} \to \mathbb{R}^{m_1 \times m_2 \times n_1 \times n_2}$ whose components are given by

$$Df_{j_1,j_2}^{i_1,i_2}(X) = \frac{\partial f_{i_2}^{i_1}}{\partial x_{j_2}^{j_1}}(X), \qquad X \in \mathbb{R}^{n_1 \times n_2}$$

where $1 \leq i_\alpha \leq m_\alpha$ and $1 \leq j_\alpha \leq n_\alpha$ are the indices, $\alpha = 1, 2$. It is easily deduced from the chain rule of ordinary calculus that if $f : \mathbb{R}^{n_1 \times n_2} \to \mathbb{R}^{m_1 \times m_2}$ and $g : \mathbb{R}^{m_1 \times m_2} \to \mathbb{R}^{l_1 \times l_2}$ are differentiable, then $g \circ f$ is differentiable with derivative $(Dg \circ f) \cdot Df : \mathbb{R}^{n_1 \times n_2} \to \mathbb{R}^{l_1 \times l_2 \times n_1 \times n_2}$, where here $\cdot$ denotes contraction over the $m_1 \times m_2$ indices. The following lemmata then follow from routine calculation.

**Lemma B.2.** *Let $bn : \mathbb{R}^{d \times N} \to \mathbb{R}^{d \times N}$ be an $\epsilon$-batchnorm layer. Then one can write $bn = v \circ m$, where $v, m : \mathbb{R}^{d \times N} \to \mathbb{R}^{d \times N}$ are given respectively by*

$$v(Y) = (N\epsilon + \|Y\|_{row}^2)^{-\frac{1}{2}} \sqrt{N} Y, \tag{25}$$

$$m(X) = X - \frac{1}{N} X 1_{N \times N}. \tag{26}$$

*One has*

$$\frac{\partial v_j^i}{\partial y_l^k} = \delta_k^i \sqrt{N} (N\epsilon + \|y^i\|^2)^{-\frac{1}{2}} \big( \delta_l^j - (N\epsilon + \|y^i\|^2)^{-1} y_l^i y_j^i \big) \tag{27}$$

*and*

$$\frac{\partial^2 v_j^i}{\partial y_n^m \partial y_l^k} = \delta_k^i \delta_m^i \sqrt{N} (N\epsilon + \|y^i\|^2)^{-\frac{3}{2}} \times$$
$$\times \big( 3(N\epsilon + \|y^i\|^2)^{-1} y_n^i y_l^i y_j^i$$
$$- (\delta_l^j y_n^i + \delta_n^l y_j^i + \delta_n^j y_l^i) \big), \tag{28}$$

*with*

$$\frac{\partial m_j^i}{\partial x_l^k} = \delta_k^i (\delta_l^j - N^{-1}). \tag{29}$$

*and all second derivatives of $m$ being zero.* □

**Lemma B.3.** *Let $wn : \mathbb{R}^{d_1 \times d_0} \to \mathbb{R}^{d_1 \times d_0}$ be an $\epsilon$-weight normalised parameterisation (Example A.1). Then one has*

$$\frac{\partial wn_j^i}{\partial w_l^k} = \delta_k^i (\epsilon + \|w^i\|^2)^{-\frac{1}{2}} \big( \delta_l^j - (\epsilon + \|w^i\|^2)^{-1} w_l^i w_j^i \big) \tag{30}$$

*and*

$$\frac{\partial wn_j^i}{\partial w_n^m \partial w_k^k} = \delta_k^i \delta_m^i (\epsilon + \|w^i\|^2)^{-\frac{3}{2}} \times$$
$$\times \big( 3(\epsilon + \|w^i\|^2)^{-1} w_n^i w_l^i w_j^i$$
$$- (\delta_l^j w_n^i + \delta_n^l w_j^i + \delta_n^j w_l^i) \big). \tag{31}$$

*Similarly, if $en : \mathbb{R}^{d_1 \times d_0} \to \mathbb{R}^{d_1 \times d_0}$ is an $\epsilon$-entry-normalised parameterisation, then*

$$\frac{\partial en_j^i}{\partial w_l^k} = \delta_k^i \delta_l^j \epsilon (\epsilon + (w_j^i)^2)^{-\frac{3}{2}} \tag{32}$$

*and*

$$\frac{\partial en_j^i}{\partial w_m^n \partial w_l^k} = -\delta_n^i \delta_m^j \delta_k^i \delta_l^j 3\epsilon (\epsilon + (w_j^i)^2)^{-\frac{3}{2}} w_j^i \tag{33}$$

*Proof of Theorem 4.5.* (1) follows from continuity of the nonlinearity and its derivative, implying boundedness of both over bounded sets in $\mathbb{R}^{d \times N}$.

(2) and (3) follow from a similar argument to the following argument for batch norm, which we give following Lemma B.2. Specifically, for the composite $f := bn \circ aff : \mathbb{R}^{d_1 \times d_0} \times \mathbb{R}^{d_0 \times N} \to \mathbb{R}^{d_1 \times N}$ defined by an $\epsilon$-BN layer and an affine layer, we will prove that over any set $B \subset \mathbb{R}^{d_0 \times N}$ consisting of matrices $X$ whose covariance matrix is nondegenerate, one has $f$, $Df$ and $Jf$ all globally bounded and Lipschitz. Indeed, $v$ (Equation (25)) is clearly globally bounded, while $Dv$ (Equation (27)) is globally bounded, decaying like $\|Y\|_{row}^{-1}$ out to infinity, and $D^2v$ (Equation (28)) is globally bounded, decaying like $\|Y\|_{row}^{-2}$ out to infinity. Consequently,

$$bn \circ aff = v \circ (m \circ aff),$$

$$D(bn \circ aff) = (Jv \circ m \circ aff) \cdot (Jm \circ aff) \cdot Daff,$$

$$J(bn \circ aff) = (Jv \circ m \circ aff) \cdot (Jm \circ aff) \cdot Jaff,$$

and similarly the derivatives of $D(\mathrm{bn} \circ \mathrm{aff})$ and $J(\mathrm{bn} \circ \mathrm{aff})$ are all globally bounded over $\mathbb{R}^{d_1 \times d_0} \times B$. The hypothesis that $B$ consist of matrices with nondegenerate covariance matrix is needed here because while $Jv \circ m \circ \mathrm{aff}$ decays like $\|A(X - \mathbb{E}[X])\|_{row}^{-1}$ out to infinity, the row-norm $\|(A(X - \mathbb{E}[X]))^i)\|^2 = (A^i(X - \mathbb{E}[X])(X - \mathbb{E}[X])^T(A^i)^T) = \|A^i\|_{\mathrm{Cov}(X)}^2$ can only be guaranteed to increase with $A$ if $\mathrm{Cov}(X)$ is nondegenerate. Thus, for instance, without the nondegeneracy hypothesis on $\mathrm{Cov}(X)$, $A \mapsto J(\mathrm{bn} \circ \mathrm{aff})(A, X)$ grows unbounded like $J\mathrm{aff}(A, X) = A \otimes \mathrm{Id}_N$ in any direction of degeneracy of $\mathrm{Cov}(X)$. Nonetheless, with the nondegenerate covariance assumption on elements of $B$, $\mathrm{bn} \circ \mathrm{aff}$ satisfies the hypotheses of Proposition 4.3 over $\mathbb{R}^{d_1 \times d_0} \times B$.

(2) and (3) now follow from essentially the same boundedness arguments as for batch norm, using Lemma B.3 in the place of Lemma B.2. However, since the row norms in this case are always defined by the usual Euclidean inner product on row-vectors, as opposed to the possibly degenerate inner product coming from the covariance matrix of the input vectors, one does not require any hypotheses aside from boundedness on the set $B$. Thus entry- and weight-normalised affine layers satisfy the hypotheses of Proposition 4.3.

Finally, (5) follows from the above arguments. More specifically, if $g : \mathbb{R}^p \times \mathbb{R}^{d \times N} \to \mathbb{R}^{d \times N}$ is any composite of layers of the above form, then $g$ satisfies the hypotheses of Proposition 4.3. Consequently, so too does the residual block $f(\theta, X) := X + g(\theta, X)$, for which $Jf(\theta, X) = \mathrm{Id}_d \otimes \mathrm{Id}_N + Jg(\theta, X)$ and $Df(\theta, X) = Dg(\theta, X)$.

$\square$

*Proof of Proposition 4.6.* In the notation of Proposition 4.2, the product $DF(\vec{\theta})DF(\vec{\theta})^T$ is the sum of the positive-semidefinite matrices $D_{\theta_l}F(\vec{\theta})D_{\theta_l}F(\vec{\theta})^T$. Therefore $\lambda(DF(\vec{\theta})) \geq \sum_l \lambda(D_{\theta_l}F(\vec{\theta}))$. The result now follows from the inequality $\lambda(AB) \geq \lambda(A)\lambda(B)$ applied inductively using Equation (11). Note that $\lambda(AB) \geq \lambda(A)\lambda(B)$ is either trivial if one or both of $A$ and $B$ have more rows than columns (in which case the right hand side is zero), and follows from the well-known inequality $\sigma(AB) \geq \sigma(A)\sigma(B)$ for the smallest singular values if both $A$ and $B$ have at least as many columns as rows. $\square$

Theorem 4.7 follows from the following two lemmata.

**Lemma B.4.** *Let* $g : \mathbb{R}^p \times \mathbb{R}^{d_0 \times N} \to \mathbb{R}^{d_1 \times N}$ *be a layer for which there exists* $\delta > 0$ *such that* $\|Jg(\theta, X)\|_2 < (1 - \delta)$ *for all* $\theta$ *and* $X$. *Let* $I : \mathbb{R}^{d_0 \times N} \to \mathbb{R}^{d_1 \times N}$ *be a linear map whose singular values are all equal to 1. Then the residual block* $f(\theta, X) := IX + g(\theta, X)$ *has* $\sigma(Jf(\theta, X)) > \delta$ *for all* $\theta$ *and* $X$.

*Proof.* Observe that

$$Jf(\theta, X) = I \otimes \mathrm{Id}_N + Jg(\theta, X). \tag{34}$$

The result then follows from Weyl's inequality: all singular values of $I \otimes \mathrm{Id}_N$ are equal to 1, so that

$$\sigma(Jf(\theta, X)) \geq 1 - \|Jg(\theta, X)\|_2 > \delta$$

for all $\theta$ and $X$. $\square$

**Lemma B.5.** *Let* $P : \mathbb{R}^p \to \mathbb{R}^{d_1 \times d_0}$ *be a parameterisation. Then*

$$\sigma(D\mathrm{aff}_P(w, X)) \geq \sigma(X)\sigma(DP(w)) \tag{35}$$

*for all* $w \in \mathbb{R}^p$ *and* $X \in \mathbb{R}^{d_0 \times N}$.

*Proof.* Follows from Equation (21) and the inequality $\sigma(AB) \geq \sigma(A)\sigma(B)$. $\square$

*Proof of Theorem 4.7.* Hypothesis 1 in Theorem 4.7 says that the residual branches of the $f_l$, $l \geq 2$, satisfy the hypotheses of Lemma B.4, so that $\sigma(Jf_l(\theta_l, f_{<l}(\vec{\theta}, X))) > 0$ for all $l \geq 2$. By the assumption that $d_{l-1} \geq d_l$, this means that $\lambda(Jf_l(\theta_l, f_{<l}(\vec{\theta}, X))) = \sigma(Jf_l(\theta_l, f_{<l}(\vec{\theta}, X)))^2 > 0$. On the other hand, hypothesis 2 together with Lemma B.5 implies that $\lambda(Df_1(\theta_1, X)) \geq \sigma(Df_1(\theta_1, X))^2 > 0$. The result now follows from Proposition 4.6. $\square$

.

*Proof of Theorem 5.1.* By Theorem 4.5, all layers satisfy the Hypotheses of Proposition 4.3 and so by Corollary 4.4, the associated loss function is globally Lipschitz, with Lipschitz constant some $\beta > 0$. Take $\eta > 0$ to be any number smaller than $2\beta^{-1}$; thus the loss can be guaranteed to be decreasing with every gradient descent step.

We now show that the network satisfies the hypotheses of Theorem 4.7. The dimension constraints in item (1) are encoded directly into the definition of the network, while the operator-norm of each of the residual branches, as products of $P(w)$ and $D\Phi$ matrices, are globally bounded by 1 by our hypotheses on these factors. For item (2), our data matrix is full-rank since it consists of linearly independent data, while by definition we have $p_1 = d_1 d_0$ with $Df_1 = D\text{aff}_{\text{en}}$ being everywhere full-rank since $\epsilon$-entry-normalisation is a diffeomorphism onto its image for any $\epsilon > 0$. Its hypotheses being satisfied by our weight-normalised residual network, Theorem 4.7 implies the parameter-function map $F$ associated to $\{f_l\}_{l=1}^L$ and $X$ satisfies $\lambda(DF(\vec{\theta})) = \sigma(DF(\vec{\theta}))^2 > 0$ for all parameters $\vec{\theta}$. There are now two cases to consider.

In the first case, the gradient descent trajectory never leaves some ball of finite radius in $\mathbb{R}^{d_1 \times d_0}$, the parameter space for the first layer. In any such ball, recalling that the first layer's parameterisation is entry-normalisation (Example A.1), the smallest singular value

$$\sigma(D\,\text{en}(w)) = \min_{1 \le i \le d_1 1 \le j \le d_0} \frac{\epsilon}{(\epsilon + (w_j^i)^2)^{\frac{3}{2}}} \tag{36}$$

of $D\,\text{en}(w)$ is uniformly lower bounded by some positive constant. Thus by Lemmas B.5 and B.4[8], the smallest singular value of $DF$ is also uniformly lower bounded by a positive constant in any such ball. It follows from Theorem 2.4 that the loss satisfies the PŁ-inequality over such a ball, so that gradient descent converges in this case at a linear rate to a global minimum.

The second and only other case that must be considered is when for each $R > 0$, there is some time $T$ for which the weight norm $\|w_t\|$ of the parameters in the first layer is greater than $R$ for all $t \ge T$. That is, the parameter trajectory in the first layer is unbounded in time. In this case, inspection of Equation (36) reveals that the smallest singular value of $DF$ *cannot* be uniformly bounded below by a positive constant over all of parameter space. Theorem 2.4 then says that there is merely a sequence $(\mu_t)_{t \in \mathbb{N}}$, with $\mu_t$ proportional to $\sigma(D\,\text{en}(w_t))$, for which

$$\ell_t - \ell^* \le \prod_{i=0}^{t} (1 - \mu_i \alpha)(\ell_0 - \ell^*), \tag{37}$$

where $\alpha = \eta(1 - 2\beta^{-1}\eta) > 0$. To guarantee convergence in this case, therefore, it suffices to show that $\prod_{t=0}^{\infty}(1 - \mu_t \alpha) = 0$; equivalently, it suffices to show that the infinite series

$$\sum_{t=0}^{\infty} \log(1 - \mu_t \alpha) \tag{38}$$

diverges.

The terms of the series (38) form a sequence of negative numbers which converges to zero. Hence, for the series (38) to diverge, it is *necessary* that $\mu_t$ decrease *sufficiently slowly* with time. By the integral test, therefore, it suffices to find an integrable function $m : [t_0, \infty) \to \mathbb{R}_{\ge 0}$ such that $\mu_t \ge m(t)$ for each integer $t \ge t_0$, for which the integral $\int_{t_0}^{\infty} \log(1 - m(t)\alpha)\,dt$ diverges.

We construct $m$ by considering the worst possible case: where each gradient descent step is in exactly the same direction going out to $\infty$, thereby decreasing $\sigma(D\,\text{en}(w))$ at the fastest possible rate. By applying an orthogonal-affine transformation to $\mathbb{R}^{d_1 d_0}$, we can assume without loss of generality that the algorithm is initialised at, and consistently steps in the direction of, the first canonical basis vector $e_1$ in $\mathbb{R}^{d_1 d_0}$. Specifically, letting $\vec{\theta}$ be the vector of parameters for all layers following the first and $w \in \mathbb{R}^{d_1 d_0}$ the first layer parameters, for $r \in \mathbb{R}_{\ge 1}$ we may assume that

$$\nabla_w \ell(\vec{\theta}, re_1) = \partial_{w_1^1} \ell(\vec{\theta}, re_1) e_1, \tag{39}$$

---

[8]See the supplementary material.

with $\partial_{w_1^1}\ell(\vec{\theta}, re_1) \geq 0$ for all $(\vec{\theta}, r)$. Let $\gamma$ denote the convex function defined by the cost $c$ (cf. Equation (6)), and let $A(\vec{\theta}, re_1)$ denote the $d_1 d_0$-dimensional row vector

$$D\gamma\big(F(\vec{\theta}, re_1)\big) \prod_{l=2}^{L} Jf_l\big(\theta_l, f_{<l}((\vec{\theta}, re_1), X)\big)(\mathrm{Id}_{d_1} \otimes X^T). \tag{40}$$

Then, in this worst possible case, the single nonzero partial derivative defining the loss gradient with respect to $w$ at the point $(\vec{\theta}, re_1)$ is given by

$$\partial_{w_1^1}\ell(\vec{\theta}, re_1)) = A(\vec{\theta}, re_1)_1 \frac{\epsilon}{(\epsilon + r^2)^{\frac{3}{2}}}, \tag{41}$$

where $A(\vec{\theta}, re_1)_1$ denotes the first component of the row vector $A(\vec{\theta}, re_1)$ (cf. Equation (21)). By Theorem 4.5, however, the magnitude of $A(\vec{\theta}, re_1)$ can be globally upper bounded by some constant $C$. Thus

$$\partial_{w_1^1}\ell(\vec{\theta}, re_1) \leq \frac{C\epsilon}{(\epsilon + r^2)^{\frac{3}{2}}} \leq \frac{C\epsilon}{r^3} \tag{42}$$

for all $\vec{\theta}$ and $r \geq 1$.

Let us therefore consider the Euler method, with step size $\eta$, applied over $\mathbb{R}_{\geq 1}$, starting from $r_0 = 1$, with respect to the vector field $V(r) = C\epsilon r^{-3}$. Labelling the iterates $(r_t)_{t \in \mathbb{N}}$, we claim that there exist constants $\gamma_1$, $\gamma_2$ and $\gamma_3$ such that $0 < r_t \leq \gamma_1 + \gamma_2(t + \gamma_3)^{\frac{1}{4}}$ for all $t \in \mathbb{N}$. Indeed, observe that the solution to the flow equation $\dot{r}(t) = C\epsilon r(t)^{-3}$ is $r(t) = (4C\epsilon t + 1)^{\frac{1}{4}}$, so the claim follows if we can show that there exists a constant $B$ such that $|r_t - r(t)| < B$ for all integer $t \geq 0$. However this follows from Theorem 10.6 of [17].

Now, since for each $t$, $r_t$ is an upper bound for the magnitude of the parameter vector $w_t \in \mathbb{R}^{d_1 d_0}$, we see from Equation (36) that the smallest singular value $\sigma(D\mathrm{en}(w_t))$ admits the lower bound

$$\sigma(D\,\mathrm{en}(w_t)) \geq \frac{\epsilon}{(\epsilon + (\gamma_1 + \gamma_2(t + \gamma_3)^{\frac{1}{4}})^2)^{\frac{3}{2}}} \tag{43}$$

for all $t \in \mathbb{N}$. Clearly, $(\epsilon + (\gamma_1 + \gamma_2(t + \gamma_3)^{\frac{1}{4}})^2)^{\frac{3}{2}} = O(t^{\frac{3}{4}})$. Hence there exist $t_0 > 0$ and $\Gamma > 0$ such that

$$\mu_t \geq \frac{\Gamma}{t^{\frac{3}{4}}} \tag{44}$$

for all integer $t \geq t_0$. Then, setting $m(t) := \Gamma t^{-\frac{3}{4}}$, the integral

$$\int_{t_0}^{\infty} \log(1 - m(t)\alpha)\, dt = \int_{t_0}^{\infty} \log\left(\frac{t^{\frac{3}{4}} - \Gamma\alpha}{t^{\frac{3}{4}}}\right) dt \tag{45}$$

diverges. It follows that gradient descent converges as $t \to \infty$ to a global minimum. $\qquad\square$

## C  Experimental details

For all our experiments, the data was standardised channel-wise using the channel-wise mean and standard deviation over the training set.

On CIFAR10/100, our models[9] were trained using SGD with a batch size of 128 and random crop/horizontal flip data augmentation. We ran 10 trials over each of the learning rates 0.2, 0.1, 0.05 and 0.02. The only exception to this is for CIFAR10 with a learning rate of 0.2, where training diverged 6 our of 10 times on the original network, so we plotted only those 4 trials where training did not diverge. Mean and standard deviation test accuracies, as well as averaged-over-final epoch loss values, are given in Tables 1 and 2.

The plots at the optimal learning rate, 0.1 for CIFAR10 and 0.05 for CIFAR100, are in Figure 3, while we provide the plots for the other learning rates in Figures 4 and 5.

---

[9]minimally modifying https://github.com/kuangliu/pytorch-cifar

Table 1: ResNet18 on CIFAR10

| | Original | | Modified | |
| Learning rate | Test accuracy | Final loss | Test accuracy | Final loss |
| --- | --- | --- | --- | --- |
| 0.20 | $48.14 \pm 19.95$ | $1.3883 \pm 0.5503$ | $62.71 \pm 6.14$ | $0.9495 \pm 0.2150$ |
| 0.10 | $91.64 \pm 0.24$ | $0.0032 \pm 0.0002$ | $91.42 \pm 0.61$ | $0.0022 \pm 0.0002$ |
| 0.05 | $91.22 \pm 0.25$ | $0.0043 \pm 0.0003$ | $91.14 \pm 0.25$ | $0.0041 \pm 0.0001$ |
| 0.02 | $89.50 \pm 0.32$ | $0.0112 \pm 0.0004$ | $88.94 \pm 0.35$ | $0.0125 \pm 0.0004$ |

Table 2: ResNet18 on CIFAR100

| | Original | | Modified | |
| Learning rate | Test accuracy | Final loss | Test accuracy | Final loss |
| --- | --- | --- | --- | --- |
| 0.20 | $69.67 \pm 0.58$ | $0.0150 \pm 0.0005$ | $70.07 \pm 0.53$ | $0.0115 \pm 0.0005$ |
| 0.10 | $70.05 \pm 0.43$ | $0.0147 \pm 0.0004$ | $70.87 \pm 0.42$ | $0.0116 \pm 0.0003$ |
| 0.05 | $70.04 \pm 0.48$ | $0.0145 \pm 0.0003$ | $70.34 \pm 0.56$ | $0.0116 \pm 0.0003$ |
| 0.02 | $69.75 \pm 0.56$ | $0.0145 \pm 0.0003$ | $70.13 \pm 0.55$ | $0.0116 \pm 0.0004$ |

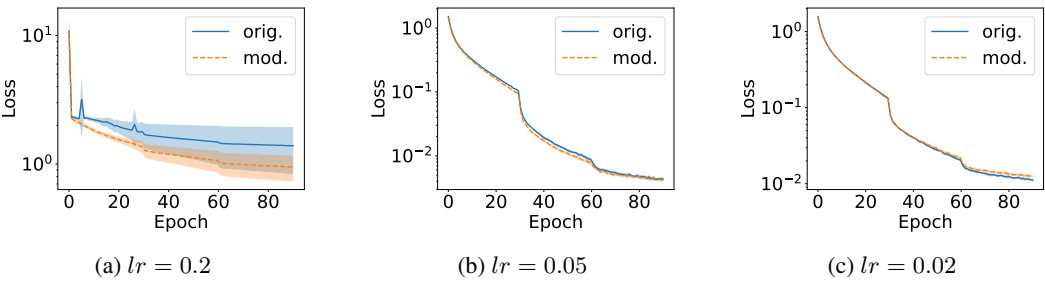

(a) $lr = 0.2$       (b) $lr = 0.05$       (c) $lr = 0.02$

Figure 4: Loss plots for ResNet18 on CIFAR10

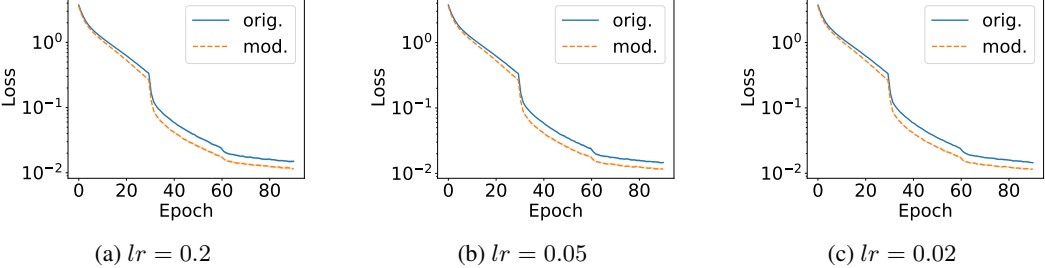

(a) $lr = 0.2$       (b) $lr = 0.05$       (c) $lr = 0.02$

Figure 5: Loss plots for ResNet18 on CIFAR100

Table 3: ResNet50 on ImageNet

| Learning rate | Original | | Modified | |
| --- | --- | --- | --- | --- |
| | Test accuracy | Final loss | Test accuracy | Final loss |
| 0.20 | $49.24 \pm 34.75$ | $1.0285 \pm 0.0066$ | – | – |
| 0.10 | $74.69 \pm 0.12$ | $0.9201 \pm 0.0014$ | $74.99 \pm 0.09$ | $0.8949 \pm 0.0019$ |
| 0.05 | $74.62 \pm 0.12$ | $0.8563 \pm 0.0041$ | $74.84 \pm 0.03$ | $0.8420 \pm 0.0030$ |
| 0.02 | $73.87 \pm 0.09$ | $0.8690 \pm 0.0025$ | $74.00 \pm 0.11$ | $0.8441 \pm 0.0026$ |

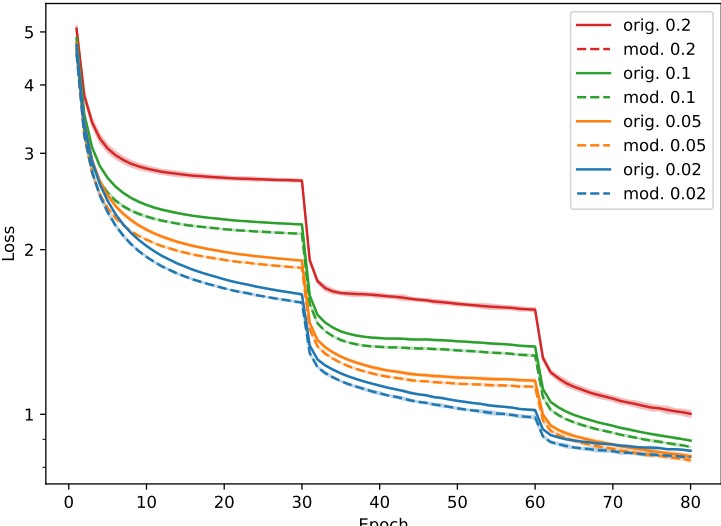

Figure 6: Loss for original and modified ResNet50 at different learning rates on ImageNet. The modified architecture did not converge at the maximum learning rate of 0.2.

On ImageNet, the models were trained using the default PyTorch ImageNet example[10], using SGD with weight decay of $1e-4$ and momentum of $0.9$, batch size of 256, and random crop/horizontal flip data augmentation. See Table 3 for the validation accuracies, and Figure 6 for the plots of the training losses at different learning rates.

---

[10]https://github.com/pytorch/examples/tree/main/imagenet

