# OpenReview forum: "On skip connections and normalisation layers in deep optimisation"
_NeurIPS.cc/2023/Conference — NeurIPS 2023 poster_

### Official Review · Reviewer_LNGS · 2023-06-13

**Soundness:** 3 good
**Presentation:** 3 good
**Contribution:** 2 fair
**Rating:** 7
**Confidence:** 3

**Summary:**

This paper studies optimisation in deep architectures, and presents a framework for theoretically capturing the role of architectural components like skip connections and normalisation layers/weight normalisation. This framework considers the curvature (smootness) and regularity properties of multi-layer networks (networks that are compositions of per-layer blocks), but differs from existing results for deep NN optimisation as it enables convergence to be shown for trajectories far away from initialisation (useful for settings when global minima only exist at infinity, like with Cross entropy loss). Finally, the authors use their theoretical results to argue that the empirically observed optimisation benefits of skip connections may be due to improving the conditioning of the loss landscape, and reinforce this argument through motivating modified skip connections which (slightly) improve performance empirically in practice.

**Strengths:**

- The paper is well motivated; it address an important question (that of bridging the gap between theory and practice in deep learning optimisation)
- The paper is well written and places itself clearly and fairly within the existing literature (in section 3)
- The paper makes contributions towards answering this question, such as: 1) building upon existing work in optimisation theory to enable global analyses that can hold for convergence far from initialisation, and 2) connecting their theoretical results to practical considerations concerning architectural design, e.g. skip connections.

**Weaknesses:**

- I'm not sure the extent to which the theoretical bounds e.g. showing the benefits of skips in terms of regularity (i.e. looking at minimum eigenvalues) translate into practice, due to the assumptions made that may not hold in practice, and also the fact that they are worst-case bounds (see lines 265-271 for the authors own words on this). I should add that this isn't really a huge criticism of this work as it applies to most works in this area.
- I'm not sure if the statement in section 6 'skip connections aid optimisation by improving loss regularity' is itself a big contribution. As the authors write in lines 138-146, there have been several works to make this statement/similar statements. Moreover, as far as I understand the theoretical arguments in Theorems 4.7 and 5.1, the arguments goes like: "Each residual layer's jacobian can be written as <identity + something bound to have eigenvalues norms less than 1>, which will have positive minimum eigenvalue, and by the chain rule their product will be too", which is quite a simple argument (forgive me if I have misunderstood, and of course it is good to formalise these things).
- The experimental results seem a bit thin, particularly I'm a bit hesistant to read too much into comparisons of training speed on ImageNet if learning rates haven't been tuned (and indeed it seems most hyperparameters were not tuned for the CIFAR results, in Appendix A.1). It would also be good to verify the empirical gains of the proposed skip architectures across different optimisers (e.g. Adam), and more depths/different residual architectures like PostAct, and also other data settings (e.g. language tasks with residual transformers), to be more convincing empirically.
- I think the Deep Kernel Shaping recent line of work (https://arxiv.org/abs/2110.01765, https://arxiv.org/abs/2203.08120) is highly relevant, and complements the current paper, so probably should be cited. Likewise, the reparameterised VGG works (https://arxiv.org/abs/2101.03697, https://arxiv.org/abs/2205.15242).

**Questions:**

- Can the authors comment on the fact that their results are designed for parameters converging to global minima at infinity, with the fact that they require bounded weights (or normalised weights/normalisation layers) in their analysis?
- Can the authors clarify what they mean by 'I is an average pool', do you pool over all pixels or only within a set filter size? It would be good just to provide a formula for this.
- Please can the authors respond to my comments in the weaknesses section. I will be happy to update my score if the rebuttal can address these.

**Update: after clarifications in the rebuttal, additional experiments, and reading the other reviews, I am increasing my score to 7**

**Limitations:**

Please see weaknesses, though the authors are quite fair about the limitations of their work and also future directions.

---

> ### Author Rebuttal · Authors · 2023-08-09
>
> We thank the reviewer for their attentive and detailed critique.
>
> *I'm not sure the extent to which the theoretical bounds…* You’re correct that the theoretical bounds are not practical. Thank you though for recognising that this is not a unique problem with our work, and is common to virtually all works in this area so far, many of which have been published in at venues such as NeurIPS and ICML.
>
> *I'm not sure if the statement in section 6…* We should have been clearer in our wording here. The “regularising effect” to which we are referring in Lines 138-146 is of a different kind to the “loss regularity” we refer to in Section 6. Specifically, by “loss regularity” in Section 6, we are referring precisely to a larger value of $\|\nabla\ell\|^2 / \ell$ (i.e. a larger PL coefficient), which therefore relates directly to optimisation theory and convergence speed. While the other works cited in Lines 138-146 study “regularising” effects of skip connections, they do not make the specific connection to loss regularity in the form of a larger PL coefficient that we do. In particular, to our knowledge no other works have yet observed the improved training speed that we predict (and observe empirically) based on this better PL regularity. We will change our wording in Lines 138-146, as well as in Section 6, to clear up this confusion.
>
> *Moreover, as far as I understand the theoretical arguments in Theorems 4.7 and 5.1…* Your understanding of the argument for Theorem 4.7 is essentially correct, and we agree that the argument here is relatively simple. However the same argument does not apply in Theorem 5.1. Theorem 5.1 already assumes that one has a positive PL coefficient everywhere (which the argument in Theorem 4.7 gives), and then proves that convergence to a global minimum is still possible even when this PL coefficient is not uniformly bounded below by a positive constant (i.e. is not a PL constant), and decays to zero as training progresses. The argument for Theorem 5.1 is significantly more involved, requiring a number of estimates on the rate of decay of the PL coefficient relative to the rate of growth in parameter norm as the trajectory tends towards a global optimum at infinity. It turns out that in order to get convergence to this optimum, it is necessary that a certain infinite series diverges. We are able to show that the PL coefficient goes to zero just slowly enough that this infinite series does indeed diverge, and asymptotic convergence of training to a global optimum is assured. The full proof can be found in the appendix.
>
> *The experimental results seem a bit thin…* To alleviate your concern, we have run the same learning rate sweep for ImageNet (to the extent our computational resources and time constraints allowed), showing that the claimed convergence improvements indeed holds across different learning rates where convergence is possible at all. We will work on having a more complete sweep done for the final version, should our paper be accepted. Please note also that we have tested networks of different depths: we used six layer networks on MNIST, ResNet18 on CIFAR and ResNet50 on ImageNet, and we used standard training schemes commonly thought of as close to optimal. Since our theory only concerns simple gradient methods, we feel that adaptive methods such as Adam are outside the scope of predictions we can reasonably make. Moreover, our Hypothesis 6.1 cannot be applied to PostAct ResNet architectures due to the presence of an additional batch norm in the skip connection. Our hypothesis could be generalised to cover this, but such a generalisation would be out of the scope of this work.
>
> *I think the Deep Kernel Shaping recent line of work…* We agree, these works are certainly relevant. We will cite them in the next version of our paper.
>
> *Can the authors comment on the fact…* In order for us to be able to guarantee Lipschitz gradients and positive PL coefficient globally, it was necessary for us to assume that all weights were normalised. One consequence of this is that the outputs of the network are bounded functions of parameters for any given training set. Restricted to this bounded set of possible outputs, the cross-entropy cost attains its global minimum at points on the boundary of this set. In particular, the global minima of cross-entropy on this bounded set are not zero. The preimages of these boundary points are always points at infinity in parameter space due to the nature of normalisation formulas: the function $x\mapsto x/\sqrt{\epsilon + x^2}$ is bounded, taking values in the interval $[-1,1]$, with the extreme values 1 and -1 only being attained at $x = \pm\infty$ respectively.
>
> *Can the authors clarify what they mean by 'I is an average pool'…* In PyTorch code, this average pool is just an nn.AvgPool2d, whose filter size and stride are both equal to the stride of the corresponding 1x1 convolution. We thus pool only over a fixed filter size. We will include more details in the next iteration of the paper.
>
> We hope we have been able to adequately address your concerns in the Weaknesses section.

---

> > ### Comment · Reviewer_LNGS · 2023-08-15
> > **Thanks, a few more questions**
> >
> > Thank you for the reply to my review. I have a few more questions which I would appreciate replies to:
> >
> > 1. In Theorem 5.1, can you provide an intuitive explanation for what $\mu_t$ correspond to? The notation is suggestive of local PL-coefficients, but as I understand it these are chosen 'hyperparameters'. Also, I think it should be $\mu_i$ not $\mu_t$ in the equation below line 256.
> > 2. What is the difference in order of LN computation in PostAct/PreAct ResNets? I am more familiar with the PostLN/PreLN difference in Transformers so I may be misunderstanding here. In particular, PreLN is the one which has the normalisation inside the residual branch, whereas PostLN uses LN on the sum of the residual branches (outside of any single branch). Is PostAct=PostLN and PreAct=PreLN in your definition? I can also check a reference/figure if you have one. It is not obvious to me why PostAct/PostLN doesn't satisfy Hypothesis 6.1
> >
> > Thanks

---

> > > ### Author Response · Authors · 2023-08-15
> > >
> > > We thank the reviewer for their additional time and attention.
> > >
> > > 1. The $\mu_t$ are indeed similar to local PL coefficients: at each step $t$, $\mu_t$ is just the smallest singular value of the derivative of the parameter-function map (see Theorem 2.4) at the parameter $\theta_t$, and is therefore a lower bound for $\|\nabla\ell(\theta_t)\|^2/\ell(\theta_t)$, which determines the speed of convergence. It is not quite true that these are hyperparameters. They are instead determined by the architecture. And it is correct that it should be $\mu_i$ and not $\mu_t$ below line 256, thank you.
> > >
> > > 2. The problem we are referring to is only an issue with the 1x1-convolutional skip-connected blocks (i.e. the blocks that reduce the feature dimension: there are 4 in each ResNet architecture). This is because we only apply our Hypothesis 6.1 to these specific blocks, and leave the others alone. In the PreAct ResNet, these 1x1-convolutional skip-connected blocks have the form:
> > > $$
> > > x\mapsto c(x)+f(x)
> > > $$
> > > where $c(x)$ is the 1x1 convolution applied to $x$, and $f(x)$ is the residual banch (a composite of BNs, ReLUs and Conv2ds) applied to $x$. On the other hand, in the PostAct (original) ResNet, the 1x1-convolutional skip-connected blocks have the form:
> > > $$
> > > x\mapsto BN(c(x)) + f(x)
> > > $$
> > > where $c(x)$ and $f(x)$ are the same as above, and BN is an additional batchnorm. Our modification is to add an additional linear map $I$, with singular values equal to 1, to the 1x1 convolution $c$. In the PreAct case this yields:
> > > $$
> > > x\mapsto I(x) + (c(x)+f(x))
> > > $$
> > > while in the PostAct case this would yield:
> > > $$
> > > x\mapsto BN(I(x) + c(x)) + f(x) \neq I(x) + (BN(c(x)) + f(x))
> > > $$
> > > In the first case, our Hypothesis predicts an upwards-shifting of the singular value distribution of the Jacobian of this block, because we have effectively added $I$, with all singular values equal to 1, to the forward pass of this block.  But due to the nonlinearity of BN, the same cannot be said for the PostAct case (note the "$\neq$" above), so that our Hypothesis does not apply.

---

> > > > ### Comment · Reviewer_LNGS · 2023-08-15
> > > > **Thanks**
> > > >
> > > > Thanks for the clarifications: I think the intuitive explanation for $\mu_i$ will help readers so would suggest including in the paper. I have increased my score to 7.

---

> > > > > ### Author Response · Authors · 2023-08-15
> > > > >
> > > > > Thank you very much for your time and consideration. We will include the intuitive explanation of $\mu_i$ in the paper.

---

### Official Review · Reviewer_9ZMW · 2023-06-26

**Soundness:** 2 fair
**Presentation:** 3 good
**Contribution:** 1 poor
**Rating:** 4
**Confidence:** 4

**Summary:**

Paper deals with convergence of deep learning making use of neural Tangent Kernels (NTK) theory and its recent advances. It is a theoretical work, claiming to provide ... "l: 7-11 Abstract: ... the only proof of which we are aware that a class of deep neural networks can be trained using gradient descent to global optima even when such optima only exist at infinity, as is the case for the cross-entropy cost." The proof, that holds for normalised networks with dense skip connections, uses an interesting approach showing that there exists sufficiently small learning rate that induces both gradient Lipschitzity and PL lower bound on gradient norm.
Interestingly the paper admits that main theorem (claiming above) is unrealistic for many reasons, i.e., more data points than input dimensions, not skipping every layer etc. The experimental results are to demonstrate that some ideas used in the main theorem could hold in practical settings as well, forming the Hypothesis 6.1.

**Strengths:**

First I'd like to thank authors for the submission of interesting paper. It has well written introduction and background knowledge sections, recalling the key building blocks (Lipschitz gradients and PL inequality) and how are they treated in related works.

Also the approach of proving the main theorem as highlighted in summary is interesting. However its 'globality' is less surprising when normalisation layers are required (than any point is weight space is translated and rescaled to lie nearby origin in probability and thus near origin NTK theory applies).


**Weaknesses:**

Impact: From my perspective and despite the nice work of proving the Main Theorem, the main weakness of this work lies in its impact. The Theorem relies on unrealistic assumptions, admitted by authors as well between lines 265-275. Then paper reduces to "just another" hypothesis why in "practice" deep learning converges to global optima.
Recall that regularisation of skip connections have been reported and theoretically argued in multitude of previous works (see, Related Work sections or Principles of riemannian geometry in neural networks, Hauser, Michael and Ray, Asok, Advances in neural information processing systems, vol. 30, 2017 just to mention one out of many uncited, for instance).

Clarity: To improve the impact the work would benefit from more accessible presentation. I leave it to authors to consider that for next iteration.

**Questions:**

Q1: The paper strives to establish new toolkit for a convergence theory in DL. One way to increase the impact and importance for ML community would be to elaborate and suggest research directions newly opened up by this approach. Could authors elaborate on open problems briefly mentioned in Discussion and suggest the ways to leverage results in the paper to advance them?

**Limitations:**

As per above, the main results relies on highly unrealistic assumptions and thus impact is limited mostly to 'proof technique' for certain problems within NTK realm. This could be powerful, but should be elaborated more, see Questions section.

---

> ### Author Rebuttal · Authors · 2023-08-09
>
> We thank the reviewer for their polite critique.
>
> We must first clear up what appear to be some misapprehensions in the reviewer’s interpretation of our work that are apparent from the Summary and Strengths section.
>
> 1.	The reviewer seems to be under the impression that we are applying NTK theory for our convergence proof. What is typically understood by “NTK theory” is the theory developed in [1], wherein it is shown that for MLPs, a certain matrix-valued function of the parameters (which, confusingly, has become known in the community as the “tangent kernel”, perhaps contributing to the reviewer's misunderstanding), converges in the infinite-width limit to a constant matrix (called the “Neural Tangent Kernel” or NTK, to which the reviewer seems to be referring) at random initialisation. **We do not invoke this theory at any time in our paper.** First, our theory concerns much more general architectures than only MLPs. Moreover, our theory does not require infinite width, nor does it require this matrix valued function (the “tangent kernel”, as opposed to the *constant* NTK) be constant (i.e. **we do not work in the NTK regime**, as the reviewer seems to think). In fact, much of the theoretical work in our paper is devoted to accounting for the variation of the smallest singular value of this matrix-valued function as training proceeds. In particular, the globality of the result does not follow from NTK theory as claimed by the reviewer.
>
> 2.	On a related note, the reviewer claims that we establish a PL lower bound on the gradient norm. This again is not accurate: as we state at Lines 107-110, when working with general cost functions such as the cross-entropy cost, a PL lower bound (in the sense of a uniform lower bound) is impossible. The best we can do is show that the quotient $\|\nabla\ell\|^2/\ell$ (to which the PL inequality gives a uniform lower bound) never vanishes. In fact, a good deal of the proof of our main theorem is spent showing how convergence to a global minimum is possible despite not having access to a uniform PL lower bound. Ours is the first work to propose such a technique.
>
> Now onto the Weaknesses:
>
> *Impact: From my perspective…* The critiques made regarding the strong assumptions could be made at least as forcefully about all other papers in the DL optimisation theory literature, over which our paper in fact makes several improvements as outlined in Lines 27-58. Many of these papers are justifiably published in outlets of similar calibre to NeurIPS, and despite strong assumptions have thousands of citations between them. It is inaccurate to say that they have had low impact. To say that our paper should be rejected based on strong assumptions alone is to say that NeurIPS shouldn’t be accepting DL optimisation theory papers at all, which we believe would be a great disservice to the field.
>
> *Recall that regularisation of skip connections have been reported…* We make no claim to be the first to have studied the effects of skip connections generally, however we do believe that we are the first to have made the link between the inclusion of skip connections and improvements to loss regularity *in the precise sense* of a larger value of $\|\nabla\ell\|^2 / \ell$. The Hauser et al paper brought up by the reviewer, while very interesting and known to us, studies skip connections from a quite different perspective which we did not believe was relevant to our work. It was for this reason that we did not cite it. If the reviewer is alleging lack of novelty then we would very much like to know what papers we have missed that are making genuinely similar claims to our own.
>
> *Q1: The paper strives to establish new toolkit for a convergence theory in DL…* Beyond the results themselves, we believe our work has two primary contributions which will be of use to others working in DL convergence theory.
>
> 1.	The framework itself. Our paper is the first we are aware of which abstracts from the any specific architectural choices, as considered in previous works, to give a general framework capable of treating all network layers on a common theoretical footing. Our framework is modular in reducing important properties of the total loss landscape to properties of the individual layers, where conditions are more easily checked, and is the first framework which allows this. Our main theorem is proof of concept of this general approach which we believe will be necessary in moving DL optimisation theory beyond the MLP examples typically considered and into more practical settings. We believe this modular approach may prove useful when trying to incorporate contributions from all layers into the estimation of PL-type bounds, which we stated as an open problem in our Discussion.
>
> 2.	Our new proof technique removes the need for a uniform PL bound. Prior to our work, all works on the optimisation of deep nets have required sufficient conditions for the optimisation trajectory to stay close to initialisation, so that a uniform PL bound computed at initialisation can be used to quantify loss decrease throughout training. These assumptions are violated for large learning rate schemes used in practice and mentioned in our Discussion. In contrast, our new proof technique is the first which permits proof of convergence without such a uniform PL bound and without insisting that the optimisation trajectory stay close to initialisation. As the only technique we are aware of that can handle such weak assumptions, we believe that it may be of utility in extending DL optimisation theory to more practical training regimes involving edge of stability.
>
> Should the paper be accepted, we will include this discussion in the extra space provided for the final iteration.
>
>
> [1] Jacot et al, Neural Tangent Kernel: Convergence and Generalization in Deep Neural Networks, NeuIPS 2018

---

> > ### Comment · Reviewer_9ZMW · 2023-08-15
> >
> > Thank you for your detailed response. Including the correcting comments on NTK and PL bounds.
> >
> > It was not claimed your were establishing $\textbf{uniform}$ bound, however. What's meant is using PL-ineq. step-wise and thus obtaining "never vanishes" for finite $T$. The point is that involving $\textit{normalisation}$ enables to apply "NTK"-like techniques step-wise (after rescaling by normalisation, which keeps the "normalized" weights closed to the origin, despite arbitrary large absolute values - the main contribution points of your approach, I guess) and taking the limit.
> >
> > Also, cited work on skip connections [Hauser] takes indeed a different geometric perspective, yet it shows that adding skip connections increases smoothness the loss landscape, and thus regularity. I agree a differential geometry perspective is quite far from approach in the manuscript but may be relevant to be cited, due to same claims.
> >
> > Not questioning the lack of novelty either. The opposite is correct, the novelty of your method has been appraised in the rebuttal. It is rather the Impact that was pointed out to be in question.
> >
> > Overall and after reading your response, I can see that one can view the work from the opposite angle. That is, including the normalisation and skip connections and thus generalising previous convergence results to more general architectures is a relevant contribution, despite these two conditions make the proofs more straightforward. Also thank you for additional comments on Q1.
> >
> > Overall, due to change of perspective above, I am ready to increase my rating by one notch to "4" giving more weight to other reviewers' opinions.

---

> > > ### Author Response · Authors · 2023-08-15
> > >
> > > We thank the reviewer for their further time and consideration. Should the paper be accepted, we will include citation and discussion of [Hauser] with the additional space provided.

---

> > > ### Author Response · Authors · 2023-08-18
> > >
> > > It seems the reviewer has not yet updated their score to 4, as they said they would.
> > >
> > > With the deadline of the discussion period arriving soon, we ask that the reviewer please update their score at their earliest convenience. We thank the reviewer again for their time.

---

> > > > ### Comment · Reviewer_9ZMW · 2023-08-18
> > > >
> > > > Thank you for reminder. Done.

---

> > > > > ### Author Response · Authors · 2023-08-18
> > > > >
> > > > > Thank you.

---

### Official Review · Reviewer_aLTj · 2023-06-30

**Soundness:** 4 excellent
**Presentation:** 4 excellent
**Contribution:** 2 fair
**Rating:** 6
**Confidence:** 4

**Summary:**

The paper provides a novel proof strategy showing the guaranteed convergence to a global optima if a particular class of deep neural networks with non-linear activation functions provided there are skip connections.  Such results have been known for linear networks so it is nice that it also applies to non-linear networks even when using a cross-entropy loss where some of the weights might diverge.  The paper goes on to show empirically that by adding in skip connections where the do not currently exist it is possible to speed up convergence of Resnets albeit marginally.

**Strengths:**

This is a very accomplished paper, clearly written by people who know what they are doing.  The paper provides a novel proof strategy to prove convergence.

**Weaknesses:**

Although technically elegant, the proof strategy only currently works in the strictly over-parameterised regime.  This is not a regime where most people work and it seems unlikely that convergence fails when we have more data-points.  I would therefore class this as a proof strategy that unfortunately doesn't really work.  Obviously, I might have misunderstood and would change my judgement if the authors could convince me that I am wrong.

The proof also seems to sacrifice linear convergence, which makes the proof slightly less interesting.  Again, maybe I am wrong about this.

More generally it is a difficult judgement call as to the value of this paper.  Clearly in deep networks people run in a regime where there is most likely many local and global optima and where they never run long enough to reach an optima (given the number of parameters, there use of mini-batches and typical size of the momentum parameter).  It is also not clear that reaching a global optimum is desirable (early stopping was a common regularisation strategy in small neural networks---it maybe unnecessary in deep neural networks because we always stop early).  In addition, Relu activation functions are usually found to give superior performance to those with Lipschitz gradients.  It thus seems that the direction of this line of research is telling us little about deep learning optimisaton.  I accept that it is interesting whether there exists a deep network with a unique minima that can be solved efficiently that demonstrates some of the power of deep learning.  This could be shown empirically, but conditions such as strict over-parameterisation makes this less useful.  I also appreciate the challenge involved in proving anything and I admire the attempt, but I still need some convincing that the direction of research is leading somewhere.

I found the empirical part of the paper slightly underwhelming.  The effect of adding a skip connection did lead to systematic improvement, but rather minor.  However, there are many explanations out there of why skip connections are beneficial and while it is plausible that the improvement is due to improved regularisation, I suspect other explanations could also be put forward (e.g. further breaking of symmetry).

This was by far the hardest paper I had to review and it took an order of magnitude longer to do it.  Even then I am at my limit of understanding so my critique might be unfounded.  If you can convince me then I am happy to change my score.

**Questions:**

Is the regime where there are more training data than free parameters physically different (i.e. we would expect in some cases we would not converge)?  If not is not the success of the proof a mathematical artefact rather than capturing the physics?

Is it the case that convergence might be sub-linear?  If so are there any guarantees on the speed of convergence?

---

> ### Author Rebuttal · Authors · 2023-08-09
>
> We thank the reviewer for their careful reading of our work and for their kind words.
>
> *Although technically elegant…* The reviewer is correct on the point that our proof technique only works in the strictly overparameterised regime. However, the reviewer is incorrect that most people do not work in this regime. This regime is common to all theoretical works on this topic that we are aware of, as well as being common to many practical settings such as image classification. All previous theoretical works also require at least one layer where the number of parameters is of at least the order of the number of training samples. There are mathematical reasons for thinking that overparameterisation is necessary for the ease of convergence of gradient descent on deep neural networks (see our response to your Question below).
>
> *The proof also seems to sacrifice linear convergence…* For strongly convex cost functions such as the square cost, our Theorem is associated (by the usual arguments for PL functions) with a linear convergence rate. It is true, however, that for non-strongly convex cost functions (such as the cross-entropy cost), linear convergence is sacrificed. However, this is to be expected since even in convex optimisation, the best-known convergence rate for fixed steps-size gradient descent on non-strongly convex objectives is sublinear: only O(1/T), with T denoting the number of iterations [1].
>
> *I still need some convincing that the direction of research is leading somewhere…* We are sympathetic with your concerns: similar concerns of our own on the limitations of existing theory motivated this work. However, it is important when evaluating the direction of a line of research to keep in mind where the line of research originated.
> It seems to us that your concerns could be directed with even greater force against the theoretical papers on the optimisation of deep nets to which our own paper is responding, many of which are published in NeurIPS, ICML and conferences of similar calibre. As a rule, these works all fall short of capturing deep learning in practice, and as outlined in Lines 25-58 the purpose of our paper is to address some of these shortcomings and bring theory closer to practice. As stated in Lines 43-58, our paper is successful in this aim. We believe it will be of interest to other theoreticians in the pursuit of more practical assumptions as well as practitioners in the principled design of architectures. Insofar as theory papers still have a place at conferences like NeurIPS, we believe that our success in our aims justifies the publication of our paper.
>
> *I found the empirical part of the paper slightly underwhelming…* We agree that the improvements were quantitatively small, but it must be remembered that we only changed four layers in each case, and these small modifications were made to architectures which are already known to be extremely easy to train (ResNets). Although the changes are small, to our knowledge no other paper in the deep learning optimisation theory literature (i.e. literature which has been concerned with proving convergence theorems) has been able to successfully make predictions about interventions on practical architectures.
> We tried to rule out other possible explanations as much as possible with our experiments, for instance by running them at multiple learning rates, and by computing all relevant mediating quantities such as singular value histograms where possible (MNIST) and showing they behaved as predicted. However, the impossibility of ruling out all other explanations is a perennial problem in science. An experiment can never prove a hypothesis true: it can only ever prove a hypothesis false.
>
> *This was by far the hardest paper I had to review…* We are sincerely grateful for your time. We know it is a difficult paper.
>
> *Is the regime where there are more training data…* We would say that this regime is mathematically different. When there are a lesser number of parameters than training data, the system of equations one is trying to solve is overdetermined (i.e. has more constraints than degrees of freedom). Such systems generically have few if any exact solutions, and whatever solutions exist tend to be isolated in locally convex regions of parameter space. In contrast, when there are many more parameters than training data the system is underdetermined (i.e. has more degrees of freedom than constraints), and typically admits infinitely many solutions occurring in large, connected regions. We recommend Figure 1 of [2] for an intuitive picture of this. Deep learning often occurs in the overparameterised regime in practice, and this overparameterisation is universally used to account for ease of optimisation in theory.
>
> *Is it the case that convergence might be sub-linear?* Our theorem gives linear convergence for the square cost, just as is achieved in comparable works. For the cross-entropy cost specifically, the convergence is indeed strictly sublinear, as is common with gradient descent on non-strongly convex objectives. We do not presently have a precise convergence rate: while such a rate would in principle be computable, since the assumptions of the theoretical part are impractical (albeit no more so than other theoretical works in this area) we felt that our time was better spent setting ourselves apart further from these other works by focussing on the more practical implications of our ideas, rather than on additional theoretical quantification.
>
> [1] Dimitri P. Bertsekas. *Convex optimization algorithms.* 2015.
>
> [2] Liu et al. *Loss landscapes and optimization in over-parameterized non-linear systems and neural networks*, arXiv:2003.00307

---

> > ### Comment · Reviewer_aLTj · 2023-08-14
> >
> > Thank you for your detailed and candid response. I am sympathetic to theoretical papers getting published in NeurIPS even when they make unrealistic assumptions (proofs are hard), although it is important that they address a real issue and are not just mathematicians talking to themselves.  It is an interesting insight that you might get converge to a unique optimum in the over-parameterised case, but not in the under-parameterised case (I was thinking about what convergence means in the wrong way).  I am not very convinced that over-parameterisation is a common situation in image applications as image augmentation is nearly always used, but I accept that some sacrifice to reality is often required.  I appreciate your attempt to break free of local regularity.  It does seem surprising that such a large class of models converges to a global optima, although I need to reflect on whether this is an important observation.  I am a bit slow so I need some time to consider if I should increase my rating of this paper.

---

> > > ### Author Response · Authors · 2023-08-15
> > >
> > > Thank you for the consideration you are giving to our paper. We agree with your comment regarding image augmentation: perhaps this is one reason why minibatching is important, so that "effective overparameterisation" is available at each step.
> > >
> > > We do not believe that we are only talking to ourselves with this paper. The point we were trying to make is that convergence analyses with more practical architectural choices (skip connections and normalisation) and the cross-entropy cost have been impossible with previously-existing techniques. Given the prevalence of these choices in practice, we felt this lack of analysis to be a real issue and worthy of our time in addressing it.
> > >
> > > Thank you again for your time.

---

> > > ### Author Response · Authors · 2023-08-21
> > >
> > > With the discussion deadline coming up soon, have you had time to reconsider your assessment?

---

### Official Review · Reviewer_kvh8 · 2023-07-06

**Soundness:** 3 good
**Presentation:** 2 fair
**Contribution:** 3 good
**Rating:** 6
**Confidence:** 3

**Summary:**

This paper proposes a formal theoretical framework to analyze gradient and convergence properties of multilayer deep networks. The authors make the following contributions: (1) they provide the first proof that certain deep networks can be trained to global optima at infinity using gradient descent; (2) the authors show that skip connections improve the loss regularity and thus improve the optimization speed; (3) they empirically show that improving the mean singular values of layer Jacobians can practically improve the convergence of ResNets on practical datasets like MNIST/CIFAR/ImageNet.

**Strengths:**

The paper has following strengths:

1. The authors present a detailed study of smoothness and regularity of deep networks with skip connections and normalization layers. By connecting these properties to singular values of layer Jacobians, authors demonstrate that skip connections improve loss regularity. This insight is interesting, and the authors verified this empirically on practical networks on MNIST, CIFAR, ImageNet datasets.

2. The proposed theoretical framework may be useful for further theoretical studies on models with skip connections.

3. The authors also conducted an interesting experiment where they augment the 1x1 convolutions with a deterministic mapping which has singular values = 1 (they used average pool that was rescaled to give singular values = 1). The authors showed that this improved training convergence as well as the final loss for all networks considered.


**Weaknesses:**

The paper has following weaknesses:

1. Some of the assumptions (as identified by the authors themselves) are not practical (lines 265-271). However, this is to be expected due to difficulty of this kind of theoretical studies. A few other assumptions include: d_{l-1} >= d_{l} (line 241) which generally does not hold (models usually have increasing number of neurons/feature maps). Also, in Theorem 5.1, can authors elaborate more on the implications of using convex cost function on the practicality of their theoretical results (line 252)?

2. Instead of using a deterministic map (avg pool rescaled to give singular values = 1), did the authors try directly using orthogonal initialization on the 1x1 convolutions in the skip connections? Would that significantly improve convergence compared to regular ResNets with default 1x1 initializations? Why did the authors use avg pool based parameterization of the skip connections?

3. There are a few important related works that the authors should discuss in section 3. Specifically, one paper defines “Layerwise Dynamical Isometry” [ICLR 2020, https://arxiv.org/abs/1906.06307] that talks about singular values of layer Jacobians and shows how improving those singular values can significantly improve training convergence of highly sparse networks. Another paper that is highly related defines NN-Mass [CVPR 2021, https://arxiv.org/abs/1910.00780] which explicitly relates singular values of layer Jacobians to skip connections. Although their theory is formulated for DenseNet-type networks, empirically, they show that their metric works for ResNets and MobileNets too. These methods show that such theory-grounded “training-free” metrics can identify efficient models directly at initialization without expensive training-based search methods. Other newer papers from this domain conduct full-blown zero-shot (training-free) Neural Architecture Search like ZenNAS (ICCV 2021, https://arxiv.org/abs/2102.01063), ZiCo (ICLR 2023, https://arxiv.org/abs/2301.11300), MAE-DET (ICML 2022, https://arxiv.org/abs/2111.13336), etc. These papers provide more insight towards expressive power or gradient properties of complex deep networks. Although the above papers may not have as much formal theory as the proposed work, it would be good to see relationship between more practical papers and the present theoretical study.


**Questions:**

Please see above.

**Limitations:**

Yes.

---

> ### Author Rebuttal · Authors · 2023-08-09
>
> We thank the reviewer for their attentive reading of our submission.
>
> *Some of the assumptions…* We agree that the assumptions are strong, although they are not exceptionally strong in comparison to other theoretical works. It is also not entirely true that models usually have an increasing number of neurons: we point to ResNets as a counterexample, which, after the very first layer, have a non-increasing number of neurons in deeper layers. The cost functions that are most frequently used in practice are the square cost and the cross entropy. Since both of these are convex we do not believe this worsens the practicality of our results.
>
> *Instead of using a deterministic map…* We did not think of trying orthogonal initialisation on the 1x1 convolutions. However based on our theory, we would predict a similar boost in training speed to what we see with our proposed modification, as we would expect it to have a similar affect on the singular values of the model, at least at initialisation. We used average pool skip connections because they aligned closely with our Hypothesis 6.1.
>
> *There are a few important related works…* Thank you very much for the recommendations. We certainly agree that these related works are relevant and will include citation of them and some discussion with the extra space provided if our paper is accepted.

---

> > ### Comment · Reviewer_kvh8 · 2023-08-19
> > **Thanks for the clarifications. I will keep my original rating.**
> >
> > I have read the author response and other reviews. I will maintain my original rating.

---

### Author Rebuttal · Authors · 2023-08-09

We refer the reviewers to the attached PDF for plots of our ImageNet experiment run at different learning rates, as requested by reviewer LNGS. Note that every trial we did of the modified network with the largest learning rate diverged. This is not surprising, given that it has been observed previously that adding skip connections increases the curvature of the loss [1]. At the other end, we see the same improvements at a learning rate of 0.05 as we predict by theory and as we observe in the 0.1 case already present in the paper. We were prevented from doing additional learning rates by time and computation constraints, and will do a more complete analysis in time for the final version if our paper is accepted.

[1] Ghorbani et al *An Investigation into Neural Net Optimization via Hessian Eigenvalue Density*, ICML 2019.

---

### Decision · Program_Chairs · 2023-09-21

**Decision:**

Accept (poster)

**Comment:**

This paper explores how normalized residual networks (potentially obtained via batch normalization and skip connections) help with optimizing deep neural networks. The authors make strong claims under specific assumptions, saying they have the first proof of convergence for training a particular class of DNNs.

All reviewers agree that the paper is analytically strong. One reviewer questions its impact, but I think introducing new training strategies and strong hypothesis is still a valid methodology. My main concern is how well the experiments back up the theory. The authors claim to be the first to prove convergence in DNN training, so they should really confirm that their optimal solution is unique and works as expected in practice. The current experiments suggest a relation to a state-of-the-art setting, which is not enough. The final accuracy obtain would not be such a big matter as it would really allow a reader to understand in which regime the theory stated in this work applies.

I would like the authors to incorporate this point, yet this despite this issue, I still recommend accepting this paper.